

# A physics-based approach to oversample multi-satellite, multi-species observations to a common grid

Kang Sun[1], Lei Zhu[2], Karen Cady-Pereira[3], Christopher Chan Miller[4], Kelly Chance[4], Lieven Clarisse[5], Pierre-François Coheur[5], Gonzalo González Abad[4], Guanyu Huang[6], Xiong Liu[4], Martin Van Damme[5], Kai Yang[7], and Mark Zondlo[8]

[1]Research and Education in eNergy, Environment and Water Institute, University at Buffalo, Buffalo, NY, USA
[2]School of Engineering and Applied Sciences, Harvard University, Cambridge, MA, USA
[3]Atmospheric and Environmental Research, Lexington, MA, USA
[4]Harvard-Smithsonian Center for Astrophysics, Cambridge, MA, USA
[5]Université libre de Bruxelles (ULB), Atmospheric Spectroscopy, Service de Chimie Quantique et Photophysique, Brussels, Belgium
[6]Department of Environmental & Health Sciences, Spelman College, Atlanta, GA, USA
[7]Department of Atmospheric and Oceanic Science, University of Maryland, College Park, MD, USA
[8]Department of Civil and Environmental Engineering, Princeton University, Princeton, NJ, USA

**Correspondence:** Kang Sun (kangsun@buffalo.edu)

**Abstract.**

Satellite remote sensing of the Earth's atmospheric composition usually samples irregularly in space and time, and many applications require spatially and temporally averaging the satellite observations (Level 2) to a regular grid (Level 3). When averaging Level 2 data over a long time period to a target Level 3 grid significantly finer than Level 2 pixels, this process is referred to as "oversampling". An agile, physics-based oversampling approach is developed to represent each satellite observation as a sensitivity distribution on the ground, instead of a point or a polygon as assumed in previous approaches. This sensitivity distribution can be determined by the spatial response function of each satellite sensor. A generalized 2-D super Gaussian function is proposed to characterize the spatial response functions of both imaging grating spectrometers (e.g., OMI, OMPS, and TROPOMI) and scanning Fourier transform spectrometers (e.g., GOSAT, IASI and CrIS). Synthetic OMI and IASI observations were generated to compare the errors due to simplifying satellite fields of view (FOV) as polygons (tessellation error) and the errors due to discretizing the smooth spatial response function on a finite grid (discretization error). The balance between these two error sources depends on the target grid resolution, the ground size of FOV, and the smoothness of spatial response functions. Explicit consideration of the spatial response function is favorable for high resolution oversampling and smoother spatial response. For OMI, it is beneficial to oversample using the spatial response functions for grid resolutions finer than ~16 km. The generalized 2-D super Gaussian function also enables smoothing of the Level 3 results by decreasing the shape-determining exponents, useful for high noise level or sparse satellite datasets. This physical oversampling is applied to OMI $NO_2$ products and IASI $NH_3$ products, showing substantially improved visualization of trace gas distribution and local gradients.




# 1 Introduction

Since the launch of the ESA Global Ozone Monitoring Experiment (GOME) in 1995, satellite observations have tremendously advanced our understanding of the processes governing the atmospheric composition, greenhouse gas emissions, and air qual-
ity (Martin, 2008; Streets et al., 2013; Jacob et al., 2016). Global distributions of atmospheric species that play critical roles in atmospheric chemistry and air pollution, such as ozone (e.g., Bak et al., 2017), $NO_2$ (e.g., Krotkov et al., 2017), $SO_2$ (e.g., Li et al., 2017a), formaldehyde (HCHO, e.g., González Abad et al., 2015), glyoxal (CHOCHO, e.g., Chan Miller et al., 2014), and BrO (e.g., Suleiman et al., 2018), have been retrieved from the backscattered solar UV-visible spectra observed by generations of polar-orbiting satellite sensors, including GOME (Burrows et al., 1999), SCIAMACHY (Bovensmann et al., 1999),
OMI (Levelt et al., 2018), GOME-2 (Munro et al., 2016), OMPS (Rodriguez et al., 2003), and TROPOMI (Veefkind et al., 2012). A constellation of geostationary satellites will provide hourly measurements of these species over North America, Europe, and Asia in the near future (Zoogman et al., 2017). Observations of the backscattered shortwave infrared solar spectra also enable the retrieval of $CO_2$, $CH_4$, and/or CO from SCIAMACHY (Buchwitz et al., 2005), GOSAT (Yoshida et al., 2011), OCO-2 (Eldering et al., 2017), and TROPOMI (Borsdorff et al., 2018; Hu et al., 2018). Moreover, many atmospheric species
have strong spectroscopic signatures in the mid-infrared and can be retrieved from the Earth's thermal emission spectra collected by satellite sensors such as MOPITT (Drummond et al., 2010), AIRS (Aumann et al., 2003), TES (Bowman et al., 2006), IASI (Clerbaux et al., 2009), and CrIS (Han et al., 2013). One species of particular significance to tropospheric chemistry and air quality is $NH_3$ (Baek et al., 2004; Paulot and Jacob, 2014), which has been successfully retrieved from TES (Shephard et al., 2011; Sun et al., 2015a), AIRS (Warner et al., 2016), IASI (Clarisse et al., 2010; Whitburn et al., 2016a; Van Damme
et al., 2017), and CrIS (Shephard and Cady-Pereira, 2015; Dammers et al., 2017).

The retrieval results from satellite sensors are usually total or partial (e.g., tropospheric or planetary boundary layer, PBL) column density at individual satellite pixels, i.e., the Level 2 product. However, the pixel geometry may vary significantly even for the same sensor (see Fig. 1 for example), and data quality screening (by cloud coverage, solar zenith angle, surface albedo, thermal contrast, etc.) often leaves only small and patchy fractions of useful Level 2 pixels for any given orbit. As such, the
Level 2 data over many orbits are often projected to a regular spatial grid to better represent the spatiotemporal variations of the target species through a gridding algorithm. These "Level 3" products help to beat down the observational noise that can be significant for individual Level 2 retrieval, make satellite data more accessible for scientific studies and the general public, and lead to additional discoveries, such as emission and lifetime estimates (Beirle et al., 2011; Valin et al., 2013; Zhu et al., 2014; de Foy et al., 2015; Fioletov et al., 2015; Whitburn et al., 2015; Liu et al., 2016; Whitburn et al., 2016b; Fioletov et al.,
2017), source identification (McLinden et al., 2012; Kort et al., 2014; McLinden et al., 2016), trend analyses (Russell et al., 2012; Lamsal et al., 2015; Duncan et al., 2016; Warner et al., 2017; Zhu et al., 2017b), assessment of environmental exposure for public health (Geddes et al., 2016; Zhu et al., 2017a), and satellite data validation (Zhu et al., 2016).



The operational Level 3 products are typically at 0.25°×0.25° or even 1°×1° resolution, which are too coarse for regional heterogeneous emission sources (e.g., urban areas), especially for species with short lifetimes. These Level 3 products are provided at fixed temporal intervals (e.g., daily, monthly, and annually). To customize the temporal interval and spatial resolution, one often needs to regrid the Level 2 data.

Various gridding algorithms have been developed to generate Level 3 maps at regional scale with much higher resolution (0.05°–0.01°) than the Level 2 pixels, and this process is generally referred to as "oversampling" (Zhu et al., 2017a, and references therein). In this work, we present an agile, physics-based oversampling approach that represents each Level 2 satellite pixel as a sensitivity distribution on the Earth's surface (e.g., the spatial response function), instead of a point or a polygon as assumed in previous methods. A generalized 2-D super Gaussian function is used to characterize the spatial response functions of both imaging grating spectrometers (e.g., OMI, OMPS, and TROPOMI) and scanning Fourier transform spectrometers (FTS, e.g., GOSAT, IASI and CrIS). Applications to multiple existing satellite datasets are also highlighted.

## 2 Satellite observations

### 2.1 OMI

The OMI instrument aboard the Aura satellite launched in 2004 is a push broom UV-visible imaging grating spectrometer. It has a daytime equatorial crossing at ∼1:42 PM local time. During normal global observation mode, the backscattered sunlight from the Earth is imaged by a telescope onto a rectangular entrance slit perpendicular to the flight direction. The light coming through the slit, which corresponds to an across-track angle of 115°, or 2600 km on the ground, is dispersed by optical gratings and mapped on two 2-D CCD detectors. Each detector image is aggregated across-track (along the length of the slit) into 60 spectra, corresponding to 60 across-track spatial pixels for the UV2 (307–383 nm) and visible (349–504 nm) bands, as shown by Fig. 1. Although the spatial response functions of OMI pixels are nonuniform (de Graaf et al., 2016; Sihler et al., 2017), the OMI pixels are widely characterized as quadrilateral polygons defined by 75% of the energy in the along-track field of view (FOV) and the half-way points of across-track FOV (the 75 FOV pixel edges from the OMPIXCOR product, Kurosu and Celarier, 2010). These OMI pixel polygons are close to rectangles, ranging from $14{\times}26$ km$^2$ at nadir to $28{\times}160$ km$^2$ at the swath edges. An alternative representation of OMI pixels is the tiled pixel edges, which produce a seamless swath image. OMI is a highly successful mission with long data records, and most of the successor missions follow a similar design (Levelt et al., 2018). The oversampling technique demonstrated here can be readily adopted for a range of OMI products and OMI's successor missions, such as OMPS, TROPOMI, and TEMPO.

### 2.2 IASI

The IASI instrument is a FTS with an across-track scanning range of 2200 km (Fig. 1). It has a local daytime equatorial crossing time of ∼9:30 AM. The first IASI instrument (IASI-A) was launched aboard the MetOp-A satellite in 2006, with the launch of IASI-B following in 2012 and the planned launch of IASI-C in near future. IASI scans across the track with 30





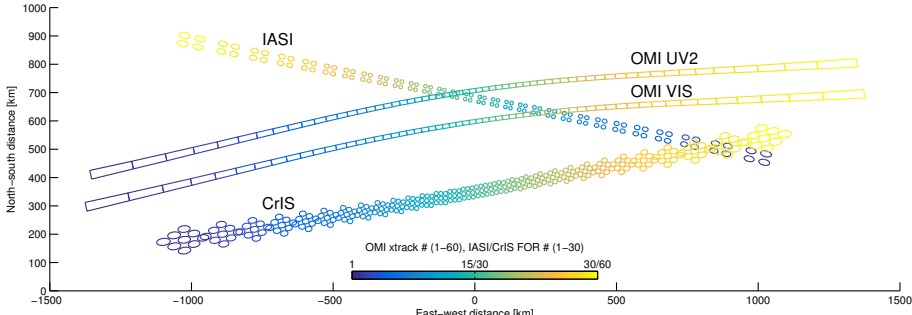

**Figure 1.** Across-track ground pixel geometry for IASI, CrIS, and the UV2/VIS (visible) bands of OMI.

mirror positions, or fields of regard (FOR), and each FOR is composed of a 2×2 array of pixels, or FOV. Each FOV projected on ground is a 12 km diameter circular footprint at nadir and elongates to ellipses towards the swath edges (Clerbaux et al., 2009). To simplify the ground pixel calculation, we represent each pixel as an ellipse with the major and minor axises and rotation angle interpolated from a look-up table based on latitude and FOR/FOV number.

5     We use the most recent neural network (NN) IASI $NH_3$ retrieval based on calculation of a hyperspectral range index (HRI) and subsequent conversion to $NH_3$ columns via a neural network (Whitburn et al., 2016a; Van Damme et al., 2017). The IASI-$NH_3$ datasets are publicly available for both IASI-A and IASI-B, with the version 2 (Van Damme et al., 2017) presenting significant improvements over version 1 (Whitburn et al., 2016a), including the negative values that are crucial for observational error averaging near the detection limit.

## 10   2.3   CrIS

The CrIS instrument, which is aboard the Suomi-NPP satellite and the series of JPSS satellites, is a step-scan FTS with 2200 km across-track width (Fig. 1). It has a daytime equatorial crossing time of ~1:30 PM local time. It has the same number of FOR as IASI, but each FOR contains 9 FOV (3×3 array), providing a better spatial coverage. Each CrIS FOV is 14 km at nadir, slightly larger than IASI. Due to the mounting angle of the scanning mirror, the FOR rotates differently at each scanning 15 angle. Similar to IASI, each CrIS pixel is represented as rotated ellipse.

    The CrIS Fast Full Physics (CFFP) $NH_3$ retrieval product is based on the TES optimal estimation approach that minimizes the differences between spectral radiances and a simulated fast forward line-by-line model (Shephard and Cady-Pereira, 2015).

## 3   Existing gridding methods

This section reviews existing gridding methods that map Level 2 data to Level 3 data. Oversampling conventionally refers to 20 the cases where Level 3 grid resolution is much smaller than the Level 2 pixel size.





### 3.1 Spatial interpolation

The spatial interpolation methods generate continuous data fields from observations made at discrete locations. The main difference between interpolation and the point- and polygon-based oversampling approaches discussed in Sect. 3.2 and Sect. 3.3 is that the values at grid cells that are not covered by satellite observations can be estimated. Therefore, the spatial interpo-
lation methods are more commonly used for satellite datasets with significant spatial gaps or requiring additional smoothing. Common spatial interpolation methods include nearest neighbors, piecewise 2-D linear interpolation, spline interpolation, and various kriging methods. The moving window block kriging method has been proposed to generate global Level 3 products for satellite observations of long-lived species, such as $CH_4$ and $CO_2$ (Tadić et al., 2015, 2017). A comprehensive review of available spatial interpolation methods for environmental variables is provided by Li and Heap (2014). There are relatively
few applications of spatial interpolation methods to regional high-resolution oversampling, where each target grid cell usually receives a large number of overlapping satellite observations. Kuhlmann et al. (2014) proposed an interpolative gridding algorithm that reconstructs the trace gas distribution by a continuous parabolic spline surface, defined on the lattice of tiled satellite pixels. This approach produces smooth regional Level 3 maps for the OMI $NO_2$ products with specifically tuned smoothing parameters, but has not been tested in non-tiled observations with significant numbers of missing values (e.g., IASI and CrIS).

### 3.2 Satellite observations as points

The simple "drop-in-the-box" gridding method can be classified into this category, as each satellite observation is assumed to be a point on the surface. The value for each target grid cell is the average of all screened satellite observations with the center of FOV falling inside the grid cell boundaries. A conventional oversampling approach has been developed based on the drop-in-the-box method; instead of only averaging "in the box", it includes satellite observations within a certain radius
(much larger than the grid size) from the center of each grid cell. This averaging radius is chosen to balance the smoothing and noise, but is also somewhat arbitrary. For example, McLinden et al. (2012) used a radius of 8 km to oversample the OMI $NO_2$ tropospheric columns and a larger radius of 24 km to oversample the OMI $SO_2$ total columns near the Canadian oil sands region; Fioletov et al. (2011) used 12 km to oversample the OMI $SO_2$ total columns over the U.S.; and Zhu et al. (2014) used 24 km to oversample the HCHO total columns near Houston, TX. This oversampling approach is referred to as "point
oversampling" hereafter, as the pixel geometry is not considered. The pixel-specific observational errors are also not taken into account.

Figure 2 reconstructs a point oversampling process for an arbitrary target grid point (red star) located near Denver, CO. OMI $NO_2$ data (Krotkov et al., 2017) over the year 2005 are used in this demonstration. Pixels with cloud fraction $\geq 30\%$ or solar zenith angle $\geq 75°$ are screened out. Only across-track positions with relatively small pixel areas (6–55 out of 1–60) are
included, a common practice to oversample OMI data. Adding pixels at the swath edges would induce more "false negative" cases, as shown below. The screened satellite pixel centers that fall within a 12-km radius (dashed circle) are plotted as black points and red triangles. The red triangles are "false positive" observations because the corresponding pixel quadrilaterals, provided by the OMPIXCOR product, do not cover the target grid point. The pixel geometry of an extreme "false positive"



case is illustrated by the pixel quadrilateral, featuring the largest separation between its boundary and the target grid point. Likewise, the "false negative" observations are plotted as purple squares, whose pixel centers fall outside the averaging circle (and hence not averaged), but these pixels cover the target grid point. An extreme case of "false negative" is also illustrated. For this example, there are 243 pixels within the 12-km radius, of which 54 are false positives (22%). There are 92 false

5   positives not included in the point oversampling. Typically, false positives are pixels closer to nadir, whereas false negatives are pixels away from nadir. In combination, the oversampled value at this grid location has contributions from a much different set of satellite observations than what should be represented. A larger averaging radius will decrease the occurrence of "false negative" cases but increase that of "false positive" cases. Because OMI pixel dimension is larger at the across-track direction, these sampling biases differ in direction; observations in the across-track direction of the target grid point are more likely to

10  become false negatives, and observations in the along-track direction are more likely to become false positives.

In reality, the OMI ground pixel footprints are not as sharp as quadrilateral boundaries (de Graaf et al., 2016), so the false positive/negative cases are not as well-defined as in Fig. 2. This will be discussed in Sect. 4.1.

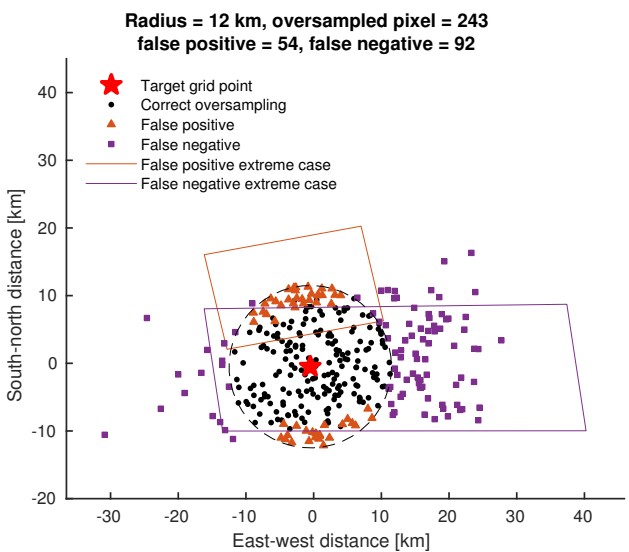

**Figure 2.** Centers of screened OMI pixels in 2005 over a target grid point (red star) near Denver, CO. OMI pixels are simplified as quadrilaterals provided by the 75FOV pixel edges from the OMPIXCOR product. Pixels that overlap with the target grid point with pixel center falling within the averaging radius (dashed circle) are plotted as black points (correct oversampling). Pixels that overlap with the target grid point with pixel center falling outside the averaging radius are plotted as purple squares (false negative). Pixels that do not overlap with the target grid point with pixel center falling in the averaging radius are plotted as red triangles (false positive). Extreme cases of false positive/negative are illustrated by OMI pixel quadrilaterals.



### 3.3 Satellite observations as polygons

This approach assumes that each satellite observation footprint is a polygon on the surface, and calculates the areal propor-
tions of grid cells inside each polygon. Because calculating these overlapping areas requires filling irregular satellite footprint
polygons with rectangular grid cells, it is also known as the "tessellation" approach. The contribution of each satellite obser-
vation to a given grid cell is weighted by the overlapping area and inversely weighted by the total pixel polygon area and the
observational uncertainty, as shown by the following equations (modified from Zhu et al., 2017a):

$$C(j) = A(j)/B(j), \tag{1}$$

where

$$A(j) = \sum_i \frac{\Omega(i)S(i,j)}{\sigma(i)^p \sum_j S(i,j)}, \tag{2}$$

$$B(j) = \sum_i \frac{S(i,j)}{\sigma(i)^p \sum_j S(i,j)}. \tag{3}$$

In the equations above, $C(j)$ is the oversampled result for destination grid cell $j$; $\Omega(i)$ is the variable to be oversampled
(e.g., total column) associated with the satellite pixel $i$; $S(i,j)$ is the overlapping area between pixel $i$ and grid cell $j$, and
hence $\sum_j S(i,j)$ is the total area of pixel $i$, assuming that the grid extends beyond all pixel boundaries. It is convenient to
normalize $S(i,j)$ by the grid cell area, leading to overlapping fractions. These equations take into account the extent of a pixel
and give more weight to a nadir observation than to an observation at the edges of the satellite swath, where the information
is more smeared out. $\sigma(i)^p$ is the uncertainty term, and the power $p$ has been assumed to be 1 (Zhu et al., 2017a) or 2 (Spurr,
2003; Van Damme et al., 2014) by different studies. If we assume each observation $\Omega(i)$ is a measurement of a constant true
value with Gaussian random error $\sigma(i)$, $p = 2$ yields the maximum likelihood estimate of the true value. However, the true
measurement and sampling errors often show heavier tails than a Gaussian distribution. In this study we adopt $p = 1$, following
Zhu et al. (2017a). The oversampled results are generally similar for both cases. Unlike the point oversampling discussed in
Sect. 3.2 where $C(j)$ is simply the average of $\Omega(i)$ within a circle, the tessellation approach fully utilizes the geometry and
error information for each satellite observation. It has been adopted by many operational Level 3 products and oversampling
studies (Xiong et al., 2006; Wenig et al., 2008; Krotkov, 2013; Van Damme et al., 2014; de Foy et al., 2015; Duncan et al.,
2016; Kim et al., 2016; Zhu et al., 2017a; Li et al., 2017b).

It is sometimes convenient to define

$$D(j) = \sum_i S(i,j) \tag{4}$$

to quantify the total number of overlapping pixel polygons used in averaging for grid cell $j$. Unlike the point oversampling,
this number does not have to be an integer due to the consideration of partial overlaps. Because the location and size of these
pixels vary day by day, averaging a large number of pixels reveals spatial patterns at scales finer than the satellite pixel scales,
if these patterns are consistent through the averaging time period.



Figure 3 illustrates the tessellation process for OMI (a) and IASI (b) pixels, where the elliptical IASI pixel is represented by a 100-vertex polygon calculated from its minor/major axises and rotational angle look-up tables. The destination grid has a resolution of 5×5 km, and the overlapping areas are normalized by the grid cell area (25 km$^2$), as labeled in each grid cell.

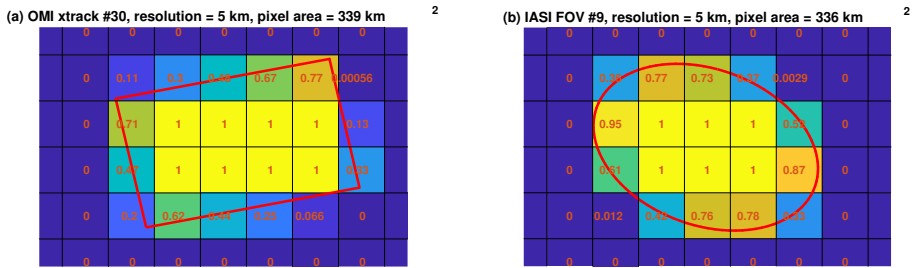

**Figure 3.** Tessellation process for OMI (a) and IASI (b) pixels. The IASI pixel is approximated by a 100-vertex polygon. Overlapping area ($S(i,j)$) between satellite pixel $i$ and grid cell $j$ is labeled at grid cell center, normalized by grid cell area (25 km$^2$).

## 4 Proposed method

### 4.1 Satellite observations as sensitivity distributions

The tessellation approach discussed in Section 3.3 inherently assumes that the satellite observation is uniformly sensitive to the scene inside the pixel polygon and has no sensitivity outside it. However, depending on target grid resolution and the spatial response function of specific satellite observations, this may be too strong of an assumption. For example, Schreier et al. (2010) characterized the complex spatial response function of the AIRS instrument and used it to improve the comparison of radiances measured by AIRS and MODIS. de Graaf et al. (2016) and Sihler et al. (2017) derived in-flight spatial response function of OMI using collocated MODIS radiance. The operational Sentinel-5 Precursor, Sentinel-5, and Sentinel-4 cloud processors also rely on the spatial response functions of the imaging grating spectrometers to accurately calculate the cloud coverage within each FOV using collocated high-resolution cloud imagers (Siddans, 2017).

For imaging grating spectrometers like OMI, the spatial response function depends on the diffraction of the fore optics, the instantaneous field of view (i.e., the instantaneous projection of the slit on the ground from the point of view of a native detector pixel), the numbers of across- and along-track bins, and the along-track movement of subsatellite point during the integration time. The satellite movement only affects the along-track direction, generally making the spatial response in the along-track direction smoother than that in the across-track direction. de Graaf et al. (2016) and Sihler et al. (2017) fitted the OMI spatial response function using a 2-D super Gaussian function to parameterize the different smoothness in the along- and across-track



directions. To standardize the representation of spatial response functions for diverse satellite sensors, we generalize the 2-D super Gaussian function as

$$S(x,y) = \exp\left(-\left(\left|\frac{x}{w_x}\right|^{k_1} + \left|\frac{y}{w_y}\right|^{k_2}\right)^{k_3}\right),$$ (5)

where

$$w_x = \frac{\text{FWHM}_x}{\ln(2)^{1/(k_1 k_3)}},$$ (6)

$$w_y = \frac{\text{FWHM}_y}{\ln(2)^{1/(k_2 k_3)}}.$$ (7)

In these equations, $x$ and $y$ are distances to the center of ground FOV in orthogonal directions, usually transformed by geometric projections of the across- and along-track directions. $\text{FWHM}_x$ and $\text{FWHM}_y$ are full widths at half maximum of the spatial response function, $S(x,y)$, in the directions of $x$ and $y$. The three exponential terms, $k_1$, $k_2$, and $k_3$, control the distribution of

spatial response, as illustrated by Fig. 4. When $k_3 = 1$ (Fig. 4a and c), Eq. 5 becomes the 2-D super Gaussian function used by de Graaf et al. (2016) and Sihler et al. (2017) to characterize the OMI spatial response:

$$S(x,y) = \exp\left(-\left|\frac{x}{w_x}\right|^{k_1} - \left|\frac{y}{w_y}\right|^{k_2}\right).$$ (8)

For OMI, $k_1 \sim 4$ and $k_2 \sim 2$ (de Graaf et al., 2016).

For FTS systems with stop-and-stare sampling, like IASI and CrIS, the spatial response function (also known as point spread

function by the community) is more simply defined by the circular aperture and some diffraction around the edge. The nadir FOV is circular with no difference between across- and along-track directions, and hence the spatial response function can be characterized by a 1-D super Gaussian function rotating around the nadir point. This rotating super Gaussian function is another special case of the generalized 2-D super Gaussian (Eq. 5) with $k_1 = k_2 = 2$ and $w_x = w_y$:

$$S(x,y) = \exp\left(-\left|\frac{R}{w}\right|^{2k_3}\right), \quad \text{where}$$

$$R = \sqrt{x^2 + y^2} \quad \text{and} \quad w = w_x = w_y.$$ (9)

The smoothness of the rotating super Gaussian is controlled by only one exponent, which equals to $2 \times k_3$. The elongated spatial response functions for off-nadir angles can be readily characterized by different values for $w_x$ and $w_y$ (Fig. 4a-b). The spatial response function of IASI is rather sharp at the edge with little variation at the top, close to a super Gaussian with an exponent of $\sim$18 (CNES, 2015). The spatial response function of CrIS is relatively smoother at the edge, best fit by a super

Gaussian with an exponent of $\sim$8 (Wang et al., 2013). Details on the spatial response functions of IASI and CrIS can be found in Appendix A.

In the generalized 2-D super Gaussian function (Eq. 5), $k_1 \times k_3$ and $k_2 \times k_3$ are the exponents in the $x$ and $y$ directions, respectively, and determine the sharpness of the spatial response in the corresponding direction. An exponent of 2 leads to a





standard Gaussian function; the larger exponents produce a top-hat shape, converging to a boxcar shape when the exponent approaches infinity (Beirle et al., 2017). Redistributing the contributions from $k_1/k_2$ and $k_3$ makes hybrid spatial response functions that may have sharp edges in sensitivity but rounded corners in space, as for the case of OMPS (Glen Jaross, personal communication). The difference between this hybrid case and conventional 2-D super Gaussian is illustrated by Fig. 4c-d.

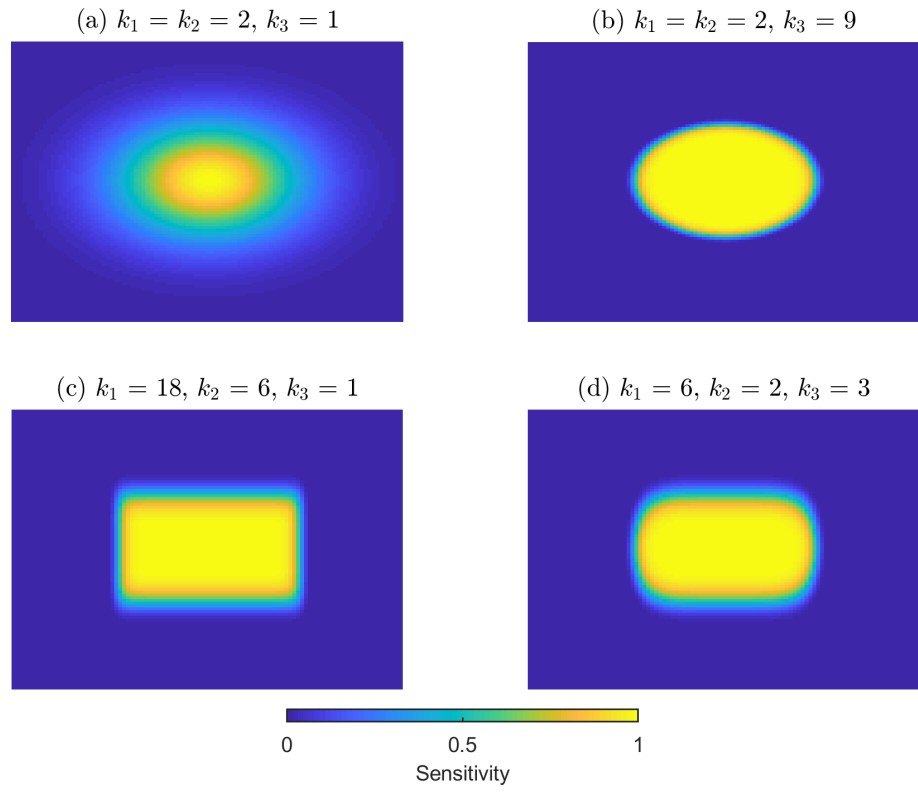

**Figure 4.** (a) Standard 2-D Gaussian function. It is both a rotating super Gaussian with an exponent of 2 and a 2-D super Gaussian function with the $x$ and $y$ direction exponents equal to 2. (b) Rotating super Gaussian with an exponent ($2 \times k_3$) of 18. (c) 2-D super Gaussian function with an exponential of 18 in the $x$ direction and an exponential of 6 in the $y$ direction. (d) A hybrid case between a rotating super Gaussian and a 2-D super Gaussian, featuring rounded corners. In all cases, $\text{FWHM}_x = 1.618 \times \text{FWHM}_y$.

5    The projection of rectangular FOV for imaging grating spectrometers like OMI on the surface at large viewing angles leads to distorted quadrilateral footprints, as shown by the polygon $ABCD$ in Fig. 5a. To account for this effect, a geometric transformation function is determined by the OMI pixel corner points ($ABCD$ in Fig. 5a) and the corresponding rectangle ($A'B'C'D'$ in Fig. 5b) defined by the distances between the middle points of opposing edges of the OMI pixel quadrilateral. The spatial response function is first calculated according to Eq. 5 with $\text{FWHM}_x = |A'D'|$, $\text{FWHM}_y = |A'B'|$ as shown in
10   Fig. 5b and then projected to match the OMI pixel corners $ABCD$ (Fig. 5a) using the geometric transformation function. This algorithm is implemented using both the OpenCV library in Python and the Image Processing Toolbox in MATLAB.





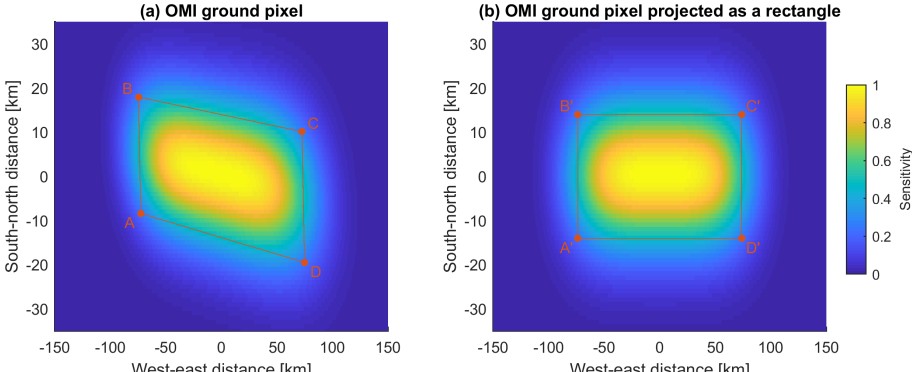

**Figure 5.** (a) OMI pixel corners ($ABCD$) for across-track position 60 out of 1–60 and spatial response function with $k_1$=4, $k_2$=2, and $k_3$=1. (b) The same OMI pixel transformed to a rectangle ($A'B'C'D'$) and the corresponding transformed spatial response function. The horizontal/vertical axises are in different scales to demonstrate that the OMI pixel is not exactly a parallelogram. As a result, the geometric transformation function is projective (not exactly affine).

The proposed oversampling approach represents each satellite observation as a sensitive distribution, instead of a point or a polygon. If the true satellite spatial response function is used as the sensitive distribution, this approach is the theoretically optimal solution to the oversampling problem, and hence referred to as "physical oversampling" hereafter. It follows the same equations as the tessellation approach as in Eq. 1-4, except that the overlapping area $S(i,j)$ is generalized to the integration of
the spatial response function of satellite observation $i$, $S(x,y|i)$, over the grid cell $j$:

$$S(i,j) = \frac{\iint_{\text{grid } j} S(x,y|i)\, \mathrm{d}x\, \mathrm{d}y}{\iint_{\text{grid } j} \mathrm{d}x\, \mathrm{d}y}, \tag{10}$$

where the denominator is the grid cell area. By normalizing the grid cell area, this accurate form of $S(i,j)$ can be directly replaced by approximating values such as $S(x,y|i)$ evaluated at the grid center. $S(i,j)/\sum_j S(i,j)$ is just the normalized spatial response function for observation $i$ so that its spatial integration is unity. If the spatial response is uniform inside the
pixel polygon and zero outside the polygon, this integration of spatial response function within the grid cell is equivalent to the fractional overlapping area used in the tessellation approach. As such, the tessellation is just the extreme case where the spatial response function is a perfect 2-D boxcar. This corresponds to $k_1 \times k_3 \rightarrow \infty$ and $k_2 \times k_3 \rightarrow \infty$ in Eq. 5.

This physical oversampling approach can also be considered as a spatial interpolation method as discussed in Sect. 3.1, because the spatial response function can be defined beyond the satellite pixel boundaries and theoretically on the entire 2-
D space. Moreover, instead of the exact form of spatial response function, the satellite observations can be represented by similar (with the same FWHM) but smoother sensitivity distributions to enhance the quality of the oversampling results. This possibility will be demonstrated in Sect. 5.2.





## 4.2 Balancing the errors from tessellation and discretization of spatial response

The tessellation approach is perfect if the spatial response of satellite observation is a boxcar, but otherwise it will introduce some error in the oversampled results (referred to as "tessellation error" hereafter). When the satellite spatial response function is smooth (instead of a boxcar), the exact solution is to calculate $S(i,j)$ as the integration of the spatial response of satellite observation $i$ over the area covered by the target grid cell $j$ (Eq. 10). Numerical integration over all grid cells may be computationally demanding and is effectively an oversampling process at a much finer grid. For a fixed grid resolution, instead of approximating the spatial response function discretely only at the grid center, we calculate a weighted average of the spatial response values at the grid center and grid corners (as proposed for MODIS by Yang and Wolfe, 2001). Because the grid corners are shared by neighboring grid cells, this approach only doubles the spatial response calculation but significantly reduces the error induced by discretization ("discretization error" hereafter). Appendix B gives a detailed comparison of tessellation and different discretization schemes.

The satellite sensors have very different spatial responses. The target grid resolution for Level 3 data ranges from 0.25° (~25 km) for many global operational products to 0.01° (~1 km) for regional oversampling. The discretization error decreases as the resolution of the target grid becomes finer and the spatial response of satellite observations becomes better resolved. At any fixed target grid resolution, spatial response functions with smoother edges are better approximated by the discretization scheme. As such, it is essential to balance the tessellation/discretization errors based on the target grid resolution and the smoothness of the satellite spatial response, so that the most accurate and efficient approximating method can be chosen.

Figure 6 compares the tessellation and discretization errors when oversampling synthetic OMI observations to a grid of 1 km (~0.01°) resolution. A checkerboard pattern is used as the "true" concentration distribution (alternating values of zeros and ones with spatial period of 20×20 km, as shown in Fig. 6a; it also shows OMI pixel polygons at across-track position #1 in red and across-track position #30 in cyan). Synthetic OMI observations are generated by sampling the checkerboard pattern using the OMI spatial response function, simplified using Eq. 8 with $k_1 = 4$, $k_2 = 2$ and discretized at very high resolution (0.05 km, or ~0.0005°) so that the spatial response distribution is always fully resolved. The locations of OMI observations are from the real OMI NO$_2$ products (Krotkov et al., 2017), filtered by cloud fraction < 25% and solar zenith angle < 75° for 2005–2006. Instead of NO$_2$ columns, the synthetic OMI observations at these locations are oversampled. The oversampled area is in the north mid-latitude (~40°N). In Fig. 6b, the oversampling is conducted at native resolution (0.05 km), and then the result is block-averaged to the 1-km target resolution to represent ideal OMI observations, as in Eq. 10. Figures 6c and e show the results for tessellation and discretization of the spatial response at 1-km resolution, where $S(i,j)$ is approximated by fractional overlapping area and the discretization scheme, respectively. They both reproduce the checkerboard pattern in general, but the tessellation method generates errors up to 40% (Fig. 6d) relative to the peak-to-trough value of the ideal observation, because the OMI spatial response is smooth (Fig. 5) instead of boxcar. In contrast, the discretization error is much smaller (Fig. 6f) because of the high resolution of the target grid (1 km).

The analysis for Fig. 6 is repeated for a range of target grid resolutions (1–50 km, or about 0.01–0.5°) and different smoothness of the spatial response functions using the same OMI observation locations. The spatial response function is assumed to





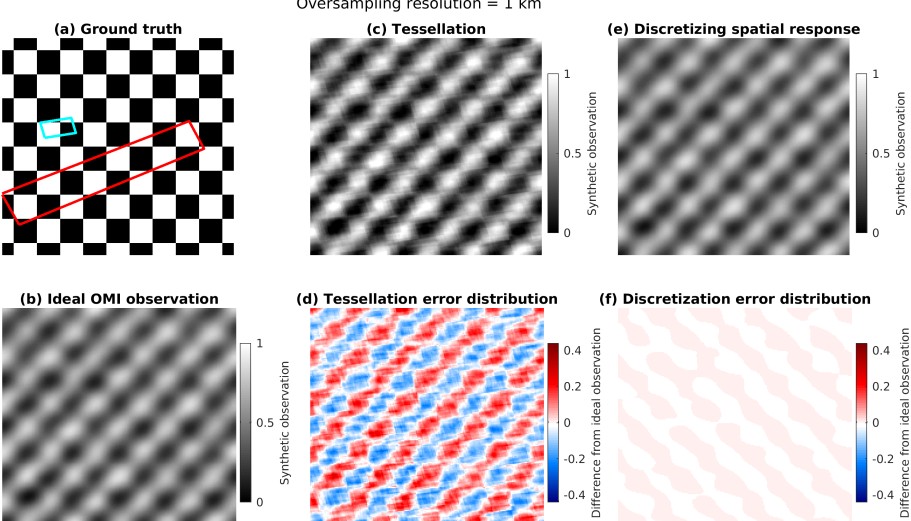

**Figure 6.** Oversampling a synthetic checkerboard pattern, shown in (a), at a spatial scale smaller than the OMI pixels to a grid resolution of 1 km. The ideal OMI observation in (b) is generated using spatial response function defined in Fig. 5 at very high resolution (0.05 km) and then coadded back to 1 km. (c) show the result from the tessellation method (assuming $S(i, j)$ equals to the overlapping area between satellite pixel $i$ and grid cell $j$). (d) shows the difference between tessellation and the ideal observation. The values in (d) are equal to (c)−(b). (e-f) show the oversampling result by discretizing the spatial response function and its difference from the ideal observation. The values in (f) are equal to (e)−(b).

be 2-D super Gaussian (Eq. 8). The exponent in the along-track direction ($k_2$) is tuned from 2 to 64, whereas the exponent in the across-track direction ($k_1$) is set to be $2 \times k_2$. Figure 7a shows, for satellite observations with quadrilateral FOV, the contour of the ratio between the discretization error and the tessellation error, calculated as the root-mean-squares of the differences between the ideal observation and the simplifications using tessellation and spatial response discretization, respectively. The

contour line of unity divides the regimes where tessellation and discretization errors are dominant: discretization of the spatial response is more accurate for high resolution oversampling of satellite observations with smooth spatial responses (small $k_1$ and $k_2$); tessellation is more accurate for coarser target resolutions and sharper spatial responses. Tessellation is perfect if $k_1$ and $k_2$ both approach infinity. The case of OMI ($k_1 = 4$, $k_2 = 2$) lies at the left edge (red vertical dashed line in Fig 7a), and its intersect with the unity contour line locates at the target resolution of ∼16 km. In other words, it is beneficial to explicitly

consider the spatial response of OMI observation for target oversampling resolutions finer than ∼16 km (about 0.15°).

Similarly, Fig. 7b shows the ratios between discretization and tessellation errors for satellite observations with circular FOVs. The pixel dimensions and locations of IASI observations for 2015–2016 are used with standard data screening, and the spatial response function is assumed to be a rotating super Gaussian (Eq. 9). The exponential term (equal to $2 \times k_3$) varies from 2 to 64. When characterizing the IASI spatial response as rotating super Gaussian function, the exponent is about 18, intersecting

the unity contour line at the target oversampling resolution of ∼2 km. If the IASI instrument had the same spatial response as



CrIS (the exponent is about 8), the intersect would be at the resolution of ∼4 km. The results would be very similar if using the CrIS observation locations instead of IASI, because the exact locations of any observations are averaged out, and the IASI and CrIS pixel sizes are similar.

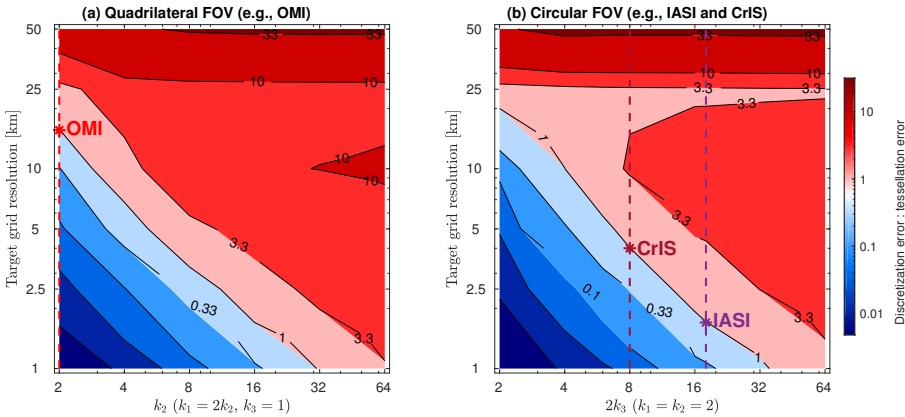

**Figure 7.** (a) The ratio between discretization/tessellation errors for different combinations of spatial response function shapes and target grid resolution. The unity contour line delineates the regime where tessellation error is larger than discretization error (blueish contours) and the regime where the discretization error is larger than tessellation error (reddish contours). The red vertical dashed line indicates the approximate spatial response for OMI. The red star marks the threshold resolution where the tessellation and discretization errors are equal for OMI. (b) Similar to (a) but the IASI pixel shapes and locations are used instead of OMI. The spatial response function exponents for CrIS and IASI and their intersects with the unity contour line are marked.

As shown by Fig. 7, the balance between tessellation and discretization errors depends on both the target grid resolution and the deviation of satellite spatial response function from an ideal 2-D boxcar shape. The uncertainty in the knowledge of the spatial response functions is not considered here, but the spatial response function can be characterized prelaunch and validated on-orbit (Schreier et al., 2010; de Graaf et al., 2016; Sihler et al., 2017). For all three cases, the tessellation error significantly outweighs the discretization error at 1 km oversampling resolution, by a factor of 4 for IASI and over 200 for OMI. Therefore, we recommend discretization of the spatial response function at 1 km (or 0.01°) resolution grid for regional scale oversampling of OMI, IASI, and CrIS data and then coadding to coarser spatial resolution if necessary. The threshold resolution where tessellation and discretization errors balance also depends on the ground size of satellite FOV. For the OMI successor missions with significantly smaller pixels (e.g., TROPOMI, TEMPO), the threshold resolution is expected to be finer.

## 5   Applications to satellite datasets

### 5.1   Physical oversampling using OMI data

Figure 8 compares the drop-in-the-box method, point oversampling, tessellation, and physical oversampling using OMI NO$_2$ tropospheric vertical column density (TVCD) within a 200×200 km square centered around a power plant in Arizona. The first



column shows the simple drop-in-the-box method on a 10-km resolution grid. The second column averages OMI observations within a 12-km radius of each grid center. These two approaches assume OMI observations as points without consideration of pixel geometry and retrieval uncertainties. The third column shows results using the tessellation approach, and the fourth column shows the physical oversampling using the OMI spatial response functions as 2-D super Gaussian function with $k_1 = 4$

and $k_2 = 2$. The target resolution is 1 km for the last three approaches. The first and third rows show the oversampled results ($C(j)$ in Eq. 1) using 5 days (1–5 July 2005) and 5 months (May–September 2005) of data, respectively. The second and fourth rows show the corresponding numbers of pixels included in the averaging for each grid cell ($D(j)$ in Eq. 4). For the drop-in-the-box approach, the total number of satellite observations included for each grid cell is much smaller and shown with a different color scale for the 5-month averaging.

The drop-in-the-box approach shows significant data gaps (5-day averaging) and high level of noise (5-month averaging), even when its target resolution is 10 times higher than the other oversampling approaches. There are two gaps where no observation is available for point oversampling over the 5 days (column 2, rows 1–2 in Fig. 8), which is an example of "false negatives" as these gaps are actually covered by OMI pixels (column 3, rows 1–2 in Fig. 8). The physical oversampling in the fourth column consistently shows the smoothest results with clear identification of the point source at the center of the

domain, because the spatial response function of OMI is properly incorporated. The oversampled $NO_2$ TVCD is biased high for the point oversampling approach because all observations within the averaging radius are averaged equally, but larger observation values generally are associated with larger uncertainties. The results from tessellation become increasingly similar to those from physical oversampling for longer averaging time, because the tessellation error is randomly distributed and will eventually be averaged out.

## 5.2 Physical oversampling using IASI data with smoother spatial sensitivity distributions

Although the physical oversampling using the true satellite spatial response functions produces the optimal estimation, the result is sometimes noisy and even unphysical, especially when the observations are noisy and sparse. In these cases, some spatial interpolation or smoothing methods are often needed. In addition to the specialized interpolation and smoothing methods discussed in Sect. 3.1, some smoothing can be applied within the oversampling framework. For example, the level of smoothing

can be adjusted by the averaging radius in the point oversampling approach. Barkley et al. (2017) used Gaussian filter to post-smooth tessellation results for OMI HCHO and CHOCHO products. When using the generalized 2-D super Gaussian function to characterize the satellite spatial response function (Eq. 5), it is also simple to tune the exponents ($k_3$ in the cases of circular FOV such as IASI/CrIS and $k_1/k_2$ in the cases of quadrilateral FOV such as OMI) so that the assumed satellite spatial sensitivity distribution is smoother than the true spatial response function. This often leads to better visualization and identification of local

hot spots, especially for products with high noise level or sparse spatial sampling. The advantage of this approach is that the smoothing is applied at the satellite pixel level (Level 2) instead of grid level (Level 3), so the geometry and error information for each satellite observation is preserved.

Figure 9 shows similar oversampling results as Fig. 8, but using IASI $NH_3$ total column density data (Van Damme et al., 2017) for 2015 in eastern Colorado, centered around a large cattle feedlot. The drop-in-the-box approach is not shown for





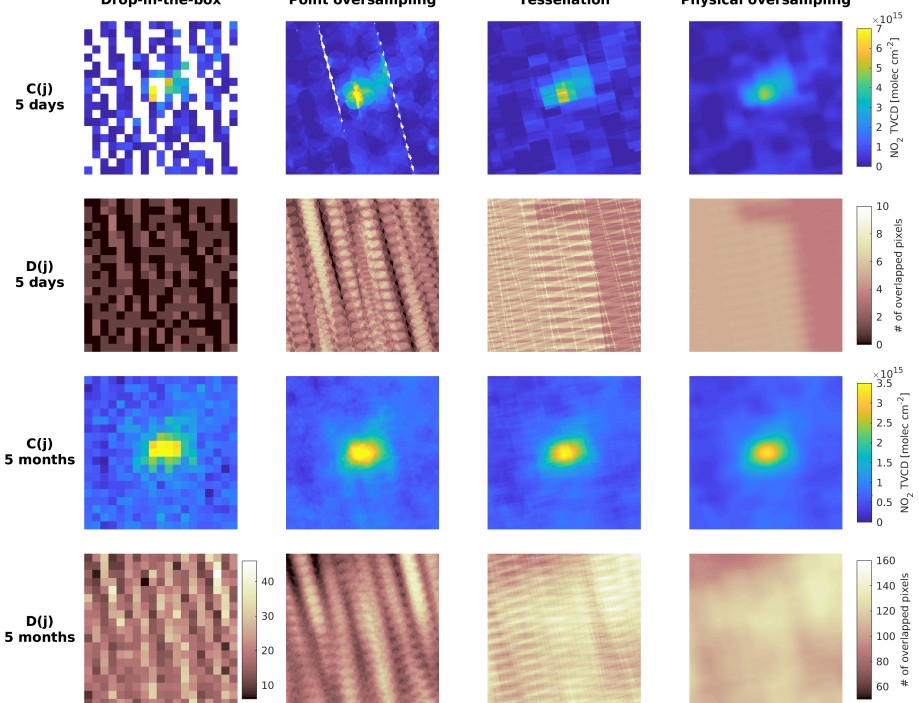

**Figure 8.** Level 3 results using drop-in-the-box method (10-km resolution, first column), point oversampling (averaging radius = 12 km, 1-km resolution, second column), tessellation (pixel corners from the OMPIXCOR product, 1-km resolution, third column), and physical oversampling (2-D super Gaussian with $k_1 = 4$ and $k_2 = 2$, 1-km resolution, fourth column). The domain size is $200 \times 200$ km. The first and third rows show the oversampled $NO_2$ TVCD for 5 days and 5 months, and the second and fourth rows show the corresponding numbers of OMI observations used in the averaging for each grid cell. Note the first panel in the fourth row is on a different color scale than the other panels in the same row.

IASI. The results from point oversampling, tessellation, and physical oversampling to a 1-km grid are presented in the first three columns. The true IASI spatial response functions have rather sharp edges (see Appendix A), so the physical oversampling shown in the third column of Fig. 9 is very similar to tessellation shown in the second column. Although this is the optimal estimation based on the physics of IASI observation, the spatial gradients are hard to identify for 5 days averaging and noisy

5    for 5 month averaging. Instead of applying smoothing after the oversampling process, the fourth column uses a smooth spatial sensitivity distribution of a 2-D standard Gaussian function ($2k_3 = 2$, rather than the true IASI spatial response function with $2k_3 \sim 18$). As illustrated by the first row in Fig. 9, the physical oversampling using smoother spatial sensitivity distributions provides the best result by clearly identifying the central point source using only sparse (5 day) data. The third row in Fig. 9 demonstrates that with 5 months of averaging, the local $NH_3$ gradients are well resolved. The point oversampling using a 12-

10    km radius overly smoothens the results, making the central hot spot artificially larger, whereas the general spatial gradients are still noisy (column 1, row 3). The overall number of IASI observations used in point oversampling is also significantly higher





than tessellation and physical oversampling, as shown by the the fourth row. This is because the 12-km averaging circle is much larger than most IASI footprints, and hence many IASI observations are double-counted as false positives. The smoothing based on physical oversampling is much more effective in suppressing the noise, and the spatial gradients are adequately preserved (column 4, row 3). This is because each satellite FOV keeps the same FWHM and overall weight, and only the distribution of sensitivity becomes more spread out.

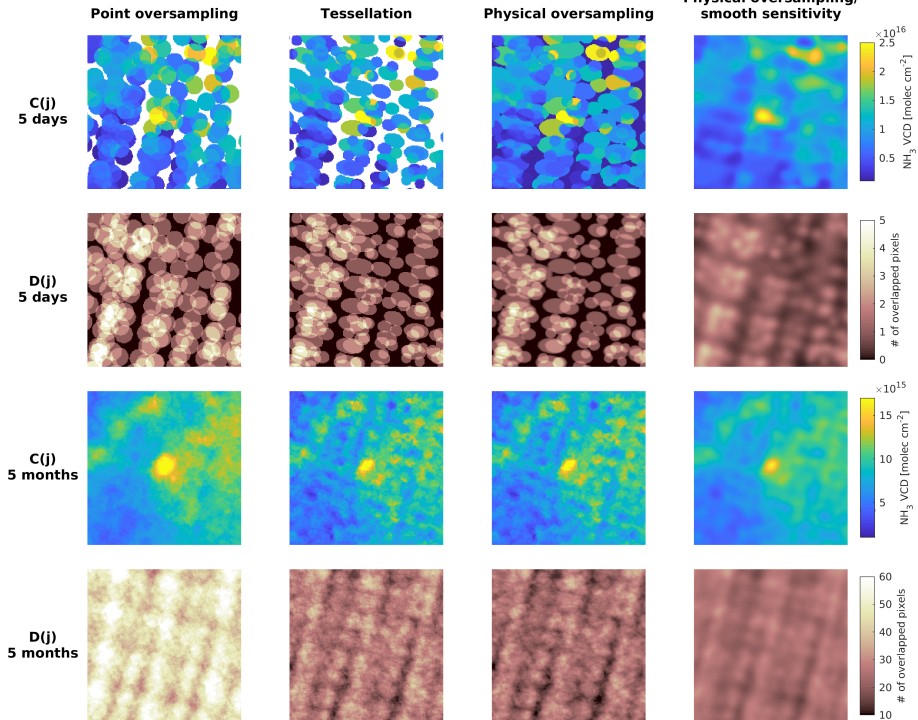

**Figure 9.** Similar to Fig. 8 using IASI NH$_3$ total column product for 2015. The drop-in-the-box approach is not included. Instead, the physical oversampling results using a smoother version of the IASI spatial response function are shown in the fourth column. The true IASI spatial response function has much sharper edges than OMI, such that the physical oversampling results (third column) are very similar to tessellation results (second column).

Oversampling based on Eq. 1-3 also provides a flexible way to categorize the results according to environmental and temporal variables. The conventional way is to save the averaging weights for each Level 2 observation (i.e., the Level 2G product, where Level 2 pixels are assigned to points of latitude/longitude grid), but the averaging weights can only be defined for a specific grid resolution. When representing each Level 2 observation as a spatial sensitivity distribution (the actual instrument spatial response function or a smoother version of it), $A(j)$ and $B(j)$ can be calculated at high spatial and temporal resolution
10 and then aggregated spatially and/or temporally. The Level 3 map $C(j)$ is just the grid-by-grid ratio of the aggregated $A(j)$ and $B(j)$. Similarly, $A(j)$ and $B(j)$ can be calculated according to environmental variables such as wind and temperature at high resolution intervals and binned to coarser categories as needed. Figure 10 shows the physical oversampling of NH$_3$



total column under southerly wind (a and c) and northerly wind (b and d) and high PBL temperature (> 15°C, a and b) and low PBL temperature (< 15°C, c and d). The average wind speed and wind direction under each category are labeled in the corresponding panels. IASI-A daytime data from 2008 to 2017 over northeastern Colorado are included in the oversampling, and a 2-D standard Gaussian is used as the spatial sensitivity distribution to smooth the results. The 3-D wind field, atmospheric

temperature, surface pressure, and PBL height are interpolated from the North American regional reanalysis (NARR; Mesinger et al., 2006) to the IASI pixel locations and overpass time. Using the concentrated animal feeding operation (CAFO) locations (colored dots; data courtesy of Daniel Bon, Colorado Department of Public Health and Environment) as a spatial reference, the downwind dispersion of total $NH_3$ column under different wind directions is clearly seen. The close match between large cattle CAFOs and the $NH_3$ hot spots seen from space confirms that they are the dominant source of atmospheric $NH_3$ in this

region. The overall abundance of $NH_3$ is significantly higher at warmer temperatures, in agreement with the previous in-situ quantification of CAFO $NH_3$ emissions in the same region (Sun et al., 2015b).

## 6   Conclusions

A conceptually simple approach is developed to oversample diverse satellite observational products to high-resolution destination grids. It represents each FOV as a sensitivity distribution on the ground, which is physically a more realistic representation

of satellite observations. This sensitivity distribution can be determined by the spatial response function of each satellite sensor. We propose a generalized 2-D super Gaussian function that can standardize the spatial response functions of many satellite sensors with distinct observation mechanisms and viewing geometries. This generalized 2-D super Gaussian function can be reduced to a rotating super Gaussian to characterize the circular FOV of IASI and CrIS, or a 2-D super Gaussian to characterize the quadrilateral FOV of OMI and its successors. It can also represent hybrid cases where the FOV is quadrilateral but with

rounded corners. When the shape-determining exponents in the generalized 2-D super Gaussian function approaches infinity, the FOV is equivalent to a polygon, as assumed in the tessellation approach.

Synthetic OMI and IASI observations were generated assuming the spatial response functions are perfectly known to compare the tessellation error and the discretization error. The balance between these two error sources depends on the target grid resolution, the ground size of FOV, and the smoothness of spatial response functions. The proposed oversampling approach

is generally more accurate for high-resolution oversampling of satellite observations with smooth spatial responses, whereas tessellation is more accurate for coarse grid resolutions and sharper spatial responses. For OMI, CrIS, and IASI, the threshold resolution where both errors equal are at ∼16 km, ∼4 km, and ∼2 km, respectively. Therefore, it is recommended to oversample to 1 km (0.01°) resolution and then coadd to coarser resolution if necessary for regional studies. The tessellation may be more desirable for generating global Level 3 products with coarse resolution. The generalized 2-D super Gaussian function

also enables smoothing of the Level 3 results by decreasing the shape-determining exponents, useful for high noise levels or sparse satellite datasets. This smoothing performed at each observation is more physically realistic than arbitrarily tuning the averaging radius and the spatial filtering of the Level 3 map as the weightings of Level 2 pixels are unchanged.





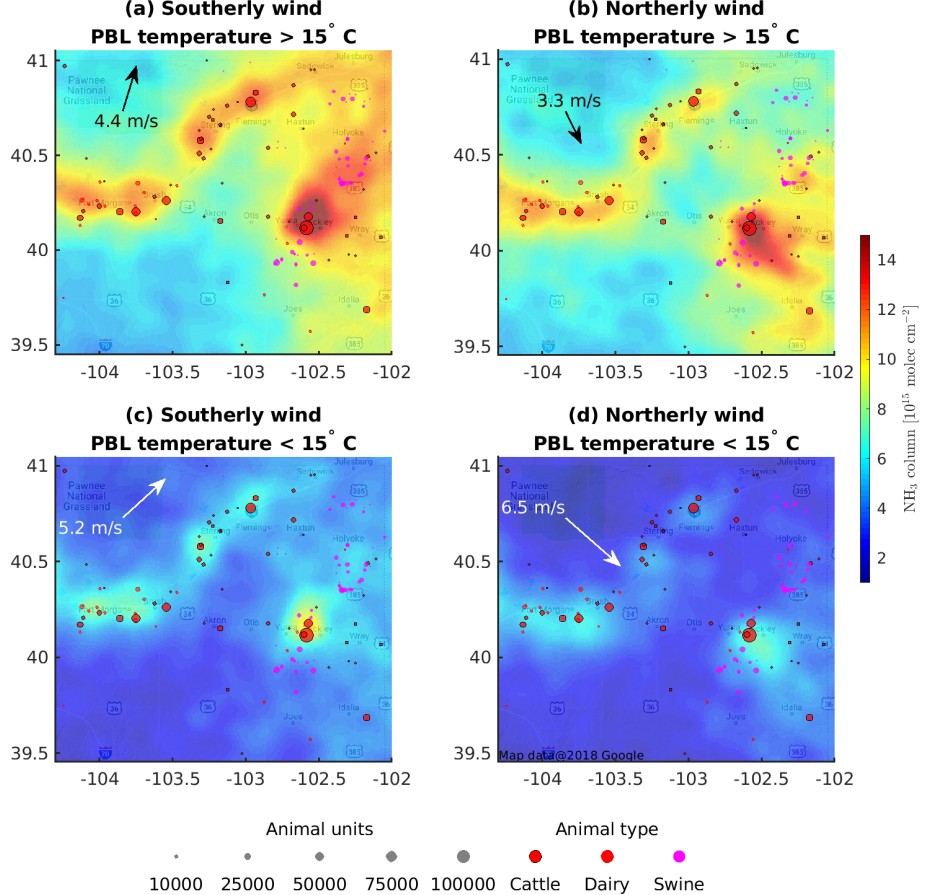

**Figure 10.** Physical oversampling results using IASI-A NH$_3$ total columns under southerly wind (a and c) and northerly wind (b and d) and high planetary boundary layer (PBL) temperature (> 15°C, a and b) and low PBL temperature (< 15°C, c and d). The text arrows show the average wind speed and wind direction at the locations/times of all IASI observations in each category. The size and location of large CAFOs are overlaid.

The new physical oversampling approach is applied to OMI NO$_2$ products and IASI NH$_3$ products, showing substantially improved visualization of trace gas distribution and local gradients. With proper consideration of the spatial response functions, this approach can be applied to multiple previous, current, and future satellite datasets, which will help to create long-term consistent data records for atmospheric composition.





## Appendix A: Spatial response functions of IASI and CrIS as rotating super Gaussian functions

The spatial response functions of IASI are tabulated at https://iasi.cnes.fr/en/IASI/A_caract_instr.htm for each of its four detector pixels. They are very close to ideal circular FOV with some smoothing at the edge and weak non-homogeneity at the top response, as shown by Fig. A1.

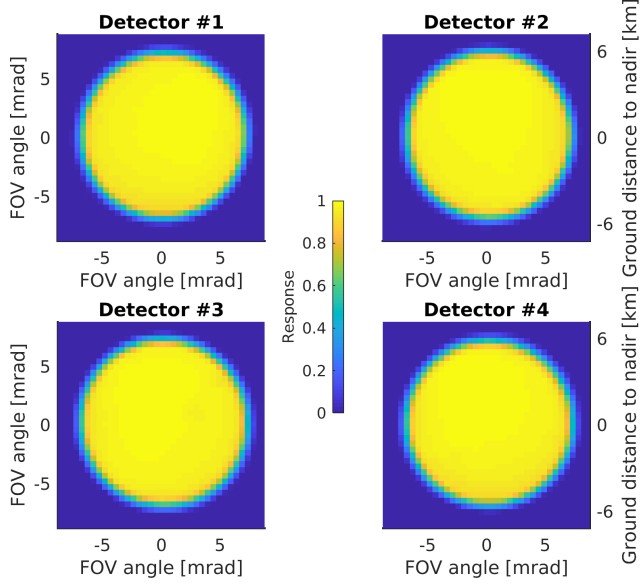

**Figure A1.** IASI spatial response functions (also known as point spread functions) defined at the viewing angular space. The corresponding ground distance at nadir is shown in the axis on the right. The IASI orbit height is assumed to be 817 km above the ground.

5    Figure A2 shows a rotating super Gaussian function (Eq. 9) fitted to the tabulated spatial response function at detector pixel #2 and the fitting residual. With only two parameters (the width and exponent of the super Gaussian), the spatial response function can be well reconstructed by the rotating super Gaussian function.

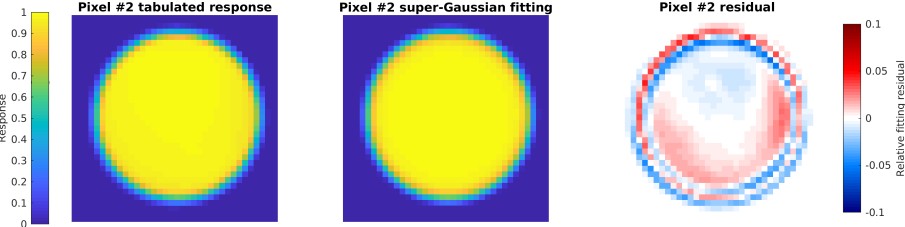

**Figure A2.** Fitting a tabulated IASI spatial response function for pixel #2 using rotating super Gaussian. The fitted exponent is 18.5.

Figure A3a shows the fitting of the across-track cross section of the spatial response function of IASI detector pixel #2 using a 1-D super Gaussian function. The FWHM is 11.6 km on the ground and the exponent is ∼18. The detailed information on




the spatial response of CrIS detectors is proprietary, but Wang et al. (2013) provides the spatial response values at a few angles, i.e., the angles of $1.2380°$, $1.1000°$, $0.9420°$, and $0.8735°$ correspond to 3%, 10%, 50%, and 70% of the peak response. Based on this information, a 1-D super Gaussian can be fitted with FWHM = 13.6 km on the ground and an exponent of 7.93, as shown by Fig. A3b. The CrIS orbit height is assumed to be 824 km above the ground.

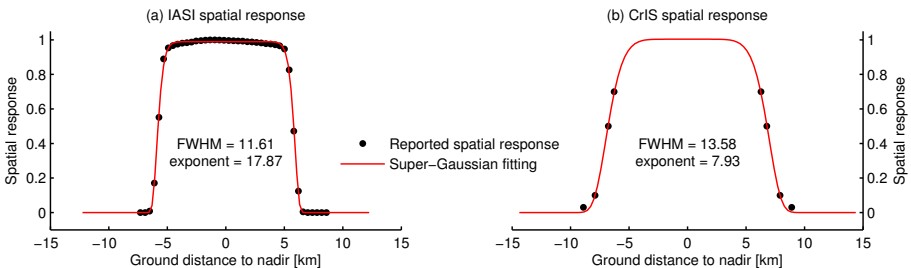

**Figure A3.** Slices of spatial response functions for IASI (a) and CrIS (b). Super Gaussian functions are fitted with the exponent $\sim 18$ for IASI and $\sim 8$ for CrIS. The spatial response functions are projected on ground to reflect actual nadir pixel sizes.

## 5 Appendix B: Comparison of discretization schemes

To compare different discretization schemes, we first construct an ideal spatial response function using OMI pixel boundaries but sharper edges ($k_1 = 12$, $k_2 = 6$, see Fig. B2a) and zoom in to a single grid cell at 5×5 km resolution (Fig. B1a). The true value of $S(i,j)$ should be the integration of the spatial response function over the grid cell area as in Eq. 10. A simple discretization scheme is to use the spatial response value at the grid center, $C$ (Fig. B1b):

$$S(i,j) = S(C), \tag{B1}$$

where $S(C)$ denotes the evaluation of continuous spatial response function $S(x,y)$ at the coordinates of the grid center $C$. A more advanced discretization scheme is to calculate the spatial response values at both the grid center and the grid corners $ABDE$ (Fig. B1c), and approximate the integration as the sum of the volumes of four triangular prisms (i.e., $ABC$, $BDC$, $DEC$, and $EAC$):

$$S(i,j) = \frac{S(A) + S(B) + S(C)}{12} + \frac{S(B) + S(D) + S(C)}{12} + \frac{S(D) + S(E) + S(C)}{12} + \frac{S(E) + S(A) + S(C)}{12}$$

$$= \frac{S(A) + S(B) + S(D) + S(E) + 2S(C)}{6}. \tag{B2}$$

Hence it is a weighted average with the weight for grid center twice of the weight for grid corners. For completeness, the assumption of tessellation is also shown in Fig. B1d, where spatial response is assumed to be unity inside the pixel boundary and zero outside. $S(i,j)$ is calculated as the fractional area covered by the portion of pixel polygon within the grid cell.

In Fig. B2, both discretization schemes and tessellation are applied to calculate $S(i,j)$ for all grid cells near the satellite FOV. Figures B2b-d show the distribution of errors from these three approximation methods, where the true $S(i,j)$ is the



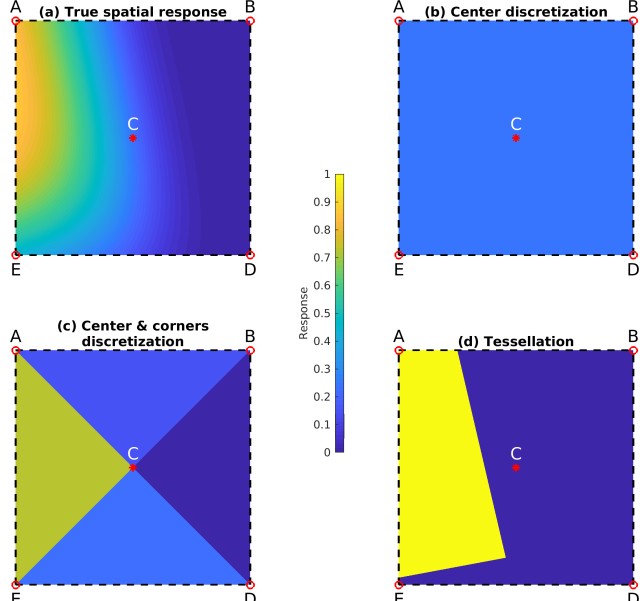

**Figure B1.** (a) An ideal spatial response function constructed using OMI across-track #30 pixel boundary and relatively sharp edges ($k_1 = 12$, $k_2 = 6$, $k_3 = 1$). Only the overlapping portion with a 5×5 km grid cell (square $ABDE$) is shown. $C$ is the grid center. (b) Simple discretization scheme, where the grid cell value is approximated by the spatial response at central position $C$. (c) The spatial response is discretized at both grid center and grid corners. See text for details. (d) Tessellation, where the spatial response is assumed to be unity inside the pixel boundary and zero outside. The polygons are color-coded by the spatial response values.

numerical integration of the high-resolution spatial response function shown in Fig. B2a. The errors in both discretization schemes (discretization only at grid center, Fig. B2b, and weighted averaging of grid center and grid corners, Fig. B2c) and the tessellation error are shown as the root-mean-square of the error distribution. The discretization scheme using both grid center and grid corner values significantly reduces the error, which in this case is also lower than the tessellation error. For a realistic OMI spatial response function ($k_1 = 4$, $k_2 = 2$), the discretization errors in both cases are significantly lower than the tessellation error at this grid resolution (5 km).

*Competing interests.* The authors declare that no competing interests.

*Acknowledgements.* We acknowledge supports from NASA's Atmospheric Composition: Aura Science Team program (sponsor contract numbers NNX14AF16G and NNX14AF56G), the RENEW Institute and School of Engineering and Applied Science at the University at Buffalo. We thank John Houck at the SAO, Thomas Kurosu at JPL, Holger Sihler at MPI-C, Glen Jaross at NASA, Rui Wang, Xuehui Guo, and Da Pan at Princeton University, and Likun Wang at University of Maryland for helpful discussions. We thank the OMI science team to





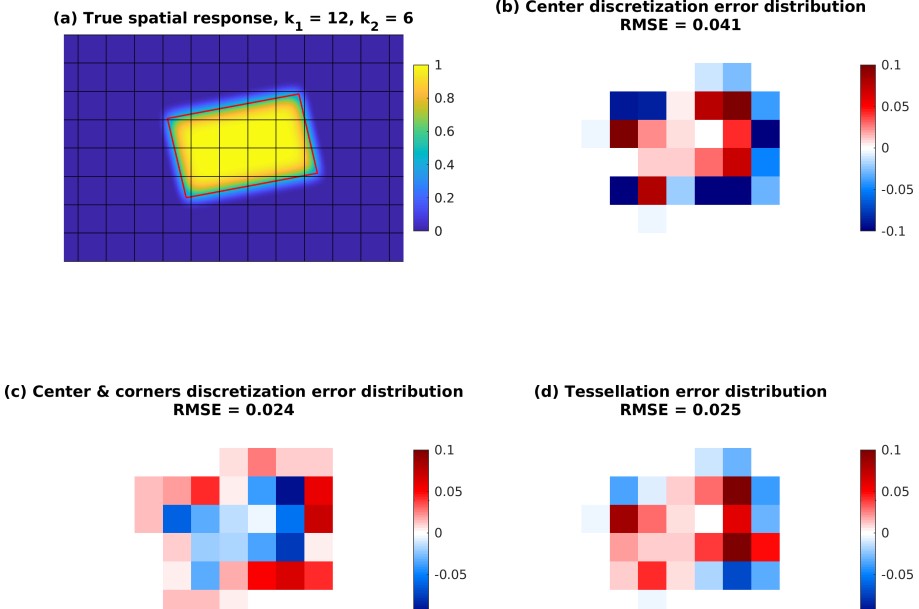

**Figure B2.** (a) An ideal spatial response function constructed using OMI across-track #30 pixel boundary (red rectangle) and relatively sharp edges ($k_1 = 12$, $k_2 = 6$, $k_3 = 1$). The destination grid at 5×5 km resolution is also shown.(b) Errors induced by discretization only at grid centers (discretized values−true values). The true value for each 5×5 km grid is calculated by numerical integration using the high-resolution spatial response shown in (a). (c) Errors induced by discretization at both grid centers and grid corners. (d) Tessellation errors. RMSE is the root-mean-square of the error distribution.

make the OMI NO$_2$ data available at https://disc.gsfc.nasa.gov/datasets/OMNO2_V003/ and the IASI science team to make the IASI NH$_3$ retrieval available at http://iasi.aeris-data.fr/NH3. L. Clarisse is a research associate with the Belgian F.R.S-FNRS and acknowledges the support.



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
