# Peer review of "A physics-based approach to oversample multi-satellite, multi-species observations to a common grid"

_Atmospheric Measurement Techniques, 2018_

## Referee Comment (RC1) · Anonymous Referee #1 · 20 Sep 2018

This paper describes a novel method to oversample satellite data. Instead of representing satellite pixels as polygons with sharp edges, the pixels are represented as 2-D Gaussian functions in which data exists beyond the edges of the polygons. As documented in previous literature (de Graaf et al., 2016), this is appropriate due to the physics of the optical measurements. The manuscript is very well-written and only minor revisions are needed.

The only glaring hole in this manuscript is a discussion on the computational time needed to complete this new oversampling technique. It is discussed at some length in Section 4.2, but it is brief and unclear. Based on my understanding, the satellite

data needs to be discretized to 0.05 km resolution? I would imagine this would be very computationally demanding. Can you provide more information on this? Can you provide some comparisons? For example in Figure 8, how long does it take for various oversampling techniques? Perhaps this should be discussed in Section 5.1.

Along the same lines, under the assumption that this technique is computationally intensive, it should be made clear in the abstract (and perhaps even title) that this technique is most useful for localized studies of air pollution and not for larger regions or global studies.

Other minor revisions and suggestions:

Page 2 Line 26: "help to beat down" is not appropriate. Suggest re-word

Page 2 Line 26: This sentence is a run-on. Suggest splitting into two sentences.

Page 3 Line 23: OMI nadir spatial resolution is typically referred to as 13 x 24 km2.

Page 3 Line 24: The sentence, "An alternative…" is unclear. Suggest re-word.

Page 6 Figure 2: This figure is a bit confusing. The second sentence in the figure caption is not necessary. Also, it would be good to provide the numbers of correct oversampled pixels (presumably 97 if I did the math correctly) and to provide all three numbers (correct, false positive, and false negative) as percentages directly on the plot and in the text.

Page 7 Line 1: The word "tessellation" should be in this section title

Page 10 Figure 4: What is the grid size of these plots? 0.05 km?

Page 12 Lines 5 -11: This section is unclear. Please re-word.

Page 13 Figure 6: It is unclear what Figures 6a and 6b are representing. What is "ground truth"? What is an "ideal OMI observation"? It's not clear what these terms mean in the context of this manuscript. The figure caption and subsequent text referring

to this figure should be clarified.

Page 18 Line 1: Are "all" southerly/northerly winds included? Or is there a threshold (e.g., only wind speeds > 1 m/s/)? Please be explicit.

Page 18 Line 1: What is "PBL temperature"? Is it the average of the temperature from the surface to the top of the PBL? Or the temperature at the top of the PBL?

Page 18 Line 5: Mention NARR spatial resolution is 32 km.

Page 18 Line 13: Remove the words "conceptually simple".

---

## Short Comment (SC1) · 25 Sep 2018

Comment on oversampling methods in general
Pertaining to "A physics-based approach to oversample multi-satellite, multi-species observations to a common grid" by K. Sun et al. (https://www.atmos-meas-tech-discuss.net/amt-2018-253/)

Levi Golston, Princeton University
Posted Aug. 25, 2018

The paper describes a new algorithm to increase spatial coverage and reduce noise at the expense of temporal resolution for satellite datasets. The super Gaussian spatial response function and comparison of tessellation and discretization errors are well described, as well as several target satellite data products.

**Main Comment**
My comment pertains to the conclusion: "This physical oversampling is applied to OMI $NO_2$ products and IASI $NH_3$ products, showing substantially improved visualization of trace gas distribution and local gradients.", as well as the premise of producing a grid that is significantly finer than the observed satellite pixels. My main argument is that the satellite measurement is the convolution of the spatial response function on the real atmospheric gas distribution, and that performing sensitivity weighted averaging like described does not change that the effective resolution of the product is still limited (to at least some extent, which does not seem to be characterized at all in the existing oversampling literature) by the Level 2 satellite pixel resolutions. Increasing resolution is actually a complex inverse problem to solve for the estimated gas distribution using the lower resolution Level 2 pixels even in the absence of noise.

By way of example, the Comment Figure below shows a simulated 1 km by 1 km region of high gas values as 'oversampled' if there was a series of 7 km by 9 km pixels each with a 2D boxcar response function. Due to the limited resolving power of the underlying pixels, the result is seen to be spread over a much larger, 13 km by 17 km area. This is a tough test case because the gas distribution is discontinuous, nevertheless the paper shows point sources applications and a very similar pattern (distinct 'circular blur') is seen in the observational results (Figs. 8, 9, and 10) at sub-pixel scales.

- If the Comment Figure is interpreted directly, then one would conclude there are gradients around the 1 km by 1 km source, when this is purely an artifact - the algorithm displays far more information than there actually is. For this reason, caution should be used in applications where there are apparent gradients around point sources, which could easily lead to mistaken conclusion about atmospheric transport or decay processes.

- Similarly, if this Comment Figure were used for a high resolution emission inventory, the emission should be proportional to the value at that specific 1 km by 1 km grid cell, which has a source strength here defined as 1. By my calculation, the corresponding oversampled grid cell has a value of only 1/63. The true source value can be retrieved - but only by integrating the whole 13 km by 17 km area which negates the high-resolution.

- One exception where higher-resolution can be used quantitatively is that that the *center* location of a source can be identified below Level 2 pixel resolution (exactly in the ideal case, but noise will of course play a role in real applications).

In summary, besides the application just mentioned, it is unclear whether the much higher-than-pixel resolution output can be justified. In the past literature, Fioletov et al. 2011 (GRL) and Fioletov et al. 2013 (JGR:A) claims 'detailed "subpixel-resolution" spatial distribution' is possible and Streets et al. 2013 (Atmos. Environ.) refers to oversampling as achieving "super-resolution". To what extent is the enhanced resolution (such as 1 km, as proposed in this manuscript) physically real? I suggest adding discussion of the limitations of the approach and caveats in interpreting the results of oversampling. This will strengthen the manuscript and help the community who may use the described algorithm or other similar methods in the future.

[Figure]

Comment Figure. (left) Real distribution of a gas with high concentration in a 1 km x 1 km area; (right) the same distribution, except as measured with a series of rectangular 7 km by 9 km satellite pixel and oversampled at 1 km resolution.

**Minor**
A point on the nomenclature of oversampling: while the term has been used in some past satellite papers, nevertheless I find it problematic since it is quite different than how it used in signal processing, where there is a well-known and widely used meaning of natively sampling at high resolution and then converting to a lower one. The authors note instead that the presented algorithm is a type of interpolation. I believe that referring to the algorithm as 'gridding' rather than 'oversampling' is formally correct and gives a much better intuition of what the algorithm actually does.

A minor point - 'agile' is also used to describe the algorithm, can a few words be added to clarify what the intended meaning is?

---

## Referee Comment (RC2) · Anonymous Referee #2 · 13 Oct 2018

This paper develops a new oversampling approach which use spatial sensitivity rather than overlapping area (tessellation approach) between target grid cell and sensor footprint as weight. The spatial sensitivity weight is calculated from sensor spatial response function which is represented by 2-D super Gaussian function. Errors from tessellation and discretization of spatial response function are well compared. Additionally, it uses several cases to show application. The paper is well written and has significant impact. I recommend its publication after addressing the following comments.

[Figure]

**Specific comments:**

1. The definition of total number of overlapping pixel polygons used in averaging for grid cell j (formula (4)) is somewhat confusing. To my understanding, $S(i,j)$ is the overlapping area, thus $D(j) = \sum_i S(i,j)$ is just sum of overlapping area and the unit of $D(j) = \sum_i S(i,j)$ is km$^2$. If so, is the unit of $D(j)$ for tessellation in figure 8 and 9 also km$^2$? Is $D(j)$ actually defined as $\sum_i \frac{S(i,j)}{\sum_j S(i,j)}$ or $\sum_i \frac{S(i,j)}{grid\ cell\ area}$ in figure 8 and 9?

2. In formula (10), why $\int \int_{grid\ j} S(x,y|i)dxdy$ is normalized by grid cell area? If $S(i,j)$ is just defined as $\int \int_{grid\ j} S(x,y|i)dxdy$, $W(i,j) = \frac{S(i,j)}{\sum_j S(i,j)}$ is the normalized spatial response function for observation i and its spatial integration $\sum_j W(i,j)$ is unity. Considering discretization of spatial response function in computation, both definitions of $S(i,j)$ are fine. Physically, should $\int \int_{grid\ j} S(x,y|i)dxdy$ be normalized by grid cell area or not?

3. Levi Golston has a good point of what is the extent of the enhanced resolution result physical real (Short comment 1 for this discussion paper, https://www.atmos-meas-tech-discuss.net/amt-2018-253/amt-2018-253-SC1-supplement.pdf). Levi Golston shows an example that "true" value is unity while oversampling result is 1/63. The result is, however, based on using 2-D boxcar spatial response function on observation generation and oversampling. If 2-D super Gaussian function is used, will it show better oversampling result? For 2-D super Gaussian function, will small $k_1$ and $k_2$ give better result than larger ones for Levi Golston's example? I suggest adding discussion of it.

---

## Author Comment (AC1) · 14 Nov 2018

Response to Referee #1:

We appreciate the very helpful feedback from the referee. The referee's comments are listed in *italics*, followed by our response in blue. New/modified text in the manuscript is in **bold**.

*1. The only glaring hole in this manuscript is a discussion on the computational time needed to complete this new oversampling technique. It is discussed at some length in Section 4.2, but it is brief and unclear. Based on my understanding, the satellite data needs to be discretized to 0.05 km resolution? I would imagine this would be very computationally demanding. Can you provide more information on this? Can you provide some comparisons? For example in Figure 8, how long does it take for various oversampling techniques? Perhaps this should be discussed in Section 5.1.*

No, the satellite data does not have to be discretized to a 0.05-km grid. The appropriate level of discretization is instrument-dependent and presented in Fig. 7. For example, the OMI oversampling grid should be discretized to a resolution finer than ~16 km, so that the physics-based oversampling can outperform the tessellation by giving smaller errors in comparison with the 0.05 km resolution case. Here the 0.05 km resolution is used as the "exact observation" where discretization error is negligible, i.e., a standard to be compared with. We do not intend to perform all oversampling at 0.05 km resolution. To clarify this point, the following sentence is added at line 27, page 12 of the original manuscript:

"**One should note that this discretization at 0.05 km is used to get the 'true' map of OMI observation where the discretization error is negligible. It is unnecessary to oversample at this fine grid in general.**"

Computational time of physical oversampling is generally not a concern. It takes $9 \times 10^{-4}$ s to run the point oversampling, and $1 \times 10^{-3}$ s to run the physical oversampling per OMI pixel for 1 km grid, which are both implemented in Matlab. Our tessellation code was written in FORTRAN, and it takes $5 \times 10^{-4}$ s per pixel for 1 km grid. Physical oversampling for elliptical pixels (IASI and CrIS) to the same grid is faster by about a factor of 4 because their pixels are smaller and no projective transformation of pixel vertices is necessary. We would like to avoid reporting exact run times in the manuscript as they will depend on the machines, programming language (FOTRAN is generally faster than Matlab/Python), and level of optimization. The bottom line is that the physical oversampling is not slower than conventional oversampling approaches. We included the following sentence at the end of Section 5.1:

"**The physical oversampling also does not require more computational resources than point oversampling and tessellation, making it suitable for a wide range of spatial scales and target grids.**"

*2. Along the same lines, under the assumption that this technique is computationally intensive, it should be made clear in the abstract (and perhaps even title) that this technique is most useful for localized studies of air pollution and not for larger regions or global studies.*

As indicated above, the physical oversampling does not require more computational resources than point oversampling and tessellation. We are currently applying the physical oversampling in

several continental-scale studies (North America, East Asia). For example, with one CPU core, it takes 15 minutes to oversample one year of CrIS data over the contiguous US (CONUS), which includes $5\times10^6$ pixels, at 2 km resolution. It takes 24 minutes to oversample to 1 km (same resolution shown in Fig. 10 of the manuscript) for the entire CONUS domain. As such, this technique is not limited to localized studies.

*Page 2 Line 26: "help to beat down" is not appropriate. Suggest re-word*

"Beat down" is revised to "**average out**".

*Page 2 Line 26: This sentence is a run-on. Suggest splitting into two sentences.*

This sentence (references are not shown) is revised to "**These 'Level 3' products help to average out the observational noise that can be significant for individual Level 2 retrieval and make satellite data more accessible for scientific studies and the general public. These products may also lead to additional discoveries, such as emission and lifetime estimates, source identification, trend analyses, assessment of environmental exposure for public health, and satellite data validation.**"

*Page 3 Line 23: OMI nadir spatial resolution is typically referred to as 13 x 24 km2.*

$13\times24$ km$^2$ is the nadir pixel size for the "tiled" OMI pixel product that assumes no overlap between adjacent pixels. However, this is not the accurate physical representation of OMI pixels because they do overlap. We clarify the numbers by adding an explanation as

"**These OMI pixel polygons are close to rectangles, ranging from $14\times26$ km$^2$ (or $13\times24$ km$^2$ if assuming non-overlapping pixels) at nadir to $28\times160$ km$^2$ at the swath edges.**"

*Page 3 Line 24: The sentence, "An alternative. . ." is unclear. Suggest re-word.*

This sentence is revised to "**Alternatively, OMI pixels can be represented as tiled polygons with no overlap between adjacent pixels. These tiled pixels produce a seamless swath image, but are less accurate, especially in the along-track direction.**"

*Page 6 Figure 2: This figure is a bit confusing. The second sentence in the figure caption is not necessary. Also, it would be good to provide the numbers of correct oversampled pixels (presumably 97 if I did the math correctly) and to provide all three numbers (correct, false positive, and false negative) as percentages directly on the plot and in the text.*

The figure and caption are revised as suggested.

*Page 7 Line 1: The word "tessellation" should be in this section title*

Revised as suggested.

*Page 10 Figure 4: What is the grid size of these plots? 0.05 km?*

These are generic 2D super Gaussian functions not following any real pixel size. The resolution is 5% of the FWHM of the vertical direction. For OMI nadir pixel this would corresponds to a resolution of 0.7 km. This information is added to the figure caption.

*Page 12 Lines 5 -11: This section is unclear. Please re-word.*

This section is revised to:

"**It is computationally demanding to numerically integrate the spatial response of all satellite pixels over each grid cell. To simplify it, one may discretize the spatial response function to the target oversampling grid and use the spatial response value at the grid center to approximate the integration. As such, the spatial response function only needs to be evaluated once per pixel per grid cell. To improve this simple discretization scheme, we calculate a weighted average of the spatial response values at the grid center and grid corners (as proposed for MODIS by Yang and Wolfe, 2001). Because the grid corners are shared by neighboring grid cells, this approach only doubles the spatial response calculation but significantly reduces the error induced by discretization ("discretization error" hereafter). Appendix B gives a detailed comparison of different discretization schemes.**"

*Page 13 Figure 6: It is unclear what Figures 6a and 6b are representing. What is "ground truth"? What is an "ideal OMI observation"? It's not clear what these terms mean in the context of this manuscript. The figure caption and subsequent text referring to this figure should be clarified.*

The text has been clarified in the response to the first comment. Moreover, the following sentences are included in the caption of Fig. 6:

"**The pattern in (a) is the ground truth of the concentration distribution. The pattern in (b) represents the ideal observation by OMI because no errors are introduced during the oversampling process.**"

*Page 18 Line 1: Are "all" southerly/northerly winds included? Or is there a threshold (e.g., only wind speeds > 1 m/s/)? Please be explicit.*

Yes, all southerly/northerly winds were included. This sentence is clarified as

"**Figure 10 shows the physical oversampling of $NH_3$ total column under southerly wind (meridional wind component > 0, a and c) and northerly wind (meridional wind component < 0, b and d) and high PBL temperature (> 15 °C, a and b) and low PBL temperature (< 15 °C, c and d). Here the PBL temperature is the average air temperature from surface to the top of the PBL, weighted by pressure.**"

*Page 18 Line 1: What is "PBL temperature"? Is it the average of the temperature from the surface to the top of the PBL? Or the temperature at the top of the PBL?*

It is the average of temperature from the surface to the top of PBL, weighted by pressure. It is clarified in the text.

*Page 18 Line 5: Mention NARR spatial resolution is 32 km.*

Done.

*Page 18 Line 13: Remove the words "conceptually simple".*

"conceptually simple" is replaced by "physics-based".

---

## Author Comment (AC2) · 14 Nov 2018

Response to Levi Golston:

We appreciate your comments (listed in *italics* hereafter). Our responses are in blue. New/modified text in the manuscript is in **bold**.

*My comment pertains to the conclusion: "This physical oversampling is applied to OMI NO2 products and IASI NH3 products, showing substantially improved visualization of trace gas distribution and local gradients.", as well as the premise of producing a grid that is significantly finer than the observed satellite pixels. My main argument is that the satellite measurement is the convolution of the spatial response function on the real atmospheric gas distribution, and that performing sensitivity weighted averaging like described does not change that the effective resolution of the product is still limited (to at least some extent, which does not seem to be characterized at all in the existing oversampling literature) by the Level 2 satellite pixel resolutions. Increasing resolution is actually a complex inverse problem to solve for the estimated gas distribution using the lower resolution Level 2 pixels even in the absence of noise.*

An important difference between resolution and sampling is brought up by these comments. This difference has been thoroughly discussed for one-dimensional data, such as satellite observed spectra, in the literature. One good example is Chance et al. (2005)[1]. The resolution of the one-dimensional spectra is constrained by the full width at half maximum (FWHM) of the Instrument Spectral Response Function, whereas the sampling is independently defined by the spectral interval of linear detector array. If the sampling interval is coarser than Nyquist sampling, the spectra are referred to as "undersampled"; if the sampling interval is (much) finer than Nyquist sampling, the spectra are referred to as "oversampled".

For two-dimensional spatial data, the resolution can be similarly constrained by the Instrument Spatial Response Function of the sensor. However, the difference between spatial resolution and the size of spatial sampling grid is often blurred. It has been a common practice to refer to the grid size of Level 3 data as "resolution" of the data. For example, "super-resolution" in the literature actually means "super fine spatial sampling", and strictly speaking, the spatial resolution stays the same as it is defined by the OMI pixel sizes that are independent of the target grid size. To avoid confusion, we replaced the term "**grid resolution**" in the manuscript by "**grid size**".

*By way of example, the Comment Figure below shows a simulated 1 km by 1 km region of high gas values as 'oversampled' if there was a series of 7 km by 9 km pixels each with a 2D boxcar response function. Due to the limited resolving power of the underlying pixels, the result is seen to be spread over a much larger, 13 km by 17 km area. This is a tough test case because the gas distribution is discontinuous, nevertheless the paper shows point sources applications and a very similar pattern (distinct 'circular blur') is seen in the observational results (Figs. 8, 9, and 10) at sub-pixel scales.*

*- If the Comment Figure is interpreted directly, then one would conclude there are gradients around the 1 km by 1 km source, when this is purely an artifact - the algorithm displays far more information than there actually is. For this reason, caution should be used in applications where*

*there are apparent gradients around point sources, which could easily lead to mistaken conclusion about atmospheric transport or decay processes.*

*- Similarly, if this Comment Figure were used for a high resolution emission inventory, the emission should be proportional to the value at that specific 1 km by 1 km grid cell, which has a source strength here defined as 1. By my calculation, the corresponding oversampled grid cell has a value of only 1/63. The true source value can be retrieved - but only by integrating the whole 13 km by 17 km area which negates the high-resolution.*

*- One exception where higher-resolution can be used quantitatively is that that the center location of a source can be identified below Level 2 pixel resolution (exactly in the ideal case, but noise will of course play a role in real applications).*

It is well-known to the community that the true spatial resolution (not the "resolution" of grid) of Level 3 maps is determined by the Level 2 pixel sizes. The apparent gradient due to finite Level 2 pixel sizes was taken into account in previous studies[2–5]. One subsection (section 4.3) and one figure (Fig. 8 of the revised manuscript) is added to clarify the definition of spatial resolution vs. spatial sampling:

**"The difference between resolution and sampling density for 1-D spectral data has been thoroughly discussed in the literature (e.g., Chance et al., 2005). However, for 2-D, spatially resolved data, it is common to refer to both the sizes of the Level 2 pixels and the size of the Level 3 grid as the spatial "resolution" of the data. To avoid confusion, it is emphasized here that the true spatial resolution is limited by the sizes of Level 2 pixels. The size of Level 3 grid only determines the density of spatial sampling, which does little to enhance the true resolving power of the data after reaching a certain point. For example, the oversampling results using synthetic OMI data at 1 km vs. 0.05 km grids are very similar (Fig. 6). Nonetheless, it is still beneficial to oversample, i.e., make Level 3 grid size significantly smaller than Level 2 pixel sizes, as demonstrated by Fig. 8. As the ground truth, an array of 2-D Gaussian functions are generated with FWHM ranging from 1 km to 16 km (the second column of Fig. 8) and peak height of unity, and this true field of concentration is measured by an imaginary sensor whose spatial response function is a 2-D super Gaussian (Eq. 8) with FWHM = 10 km and $k_1 = k_2 = 8$ (the first column and the white boxes inserted in the third column). The third column shows the oversampling results using 10000 randomly located observations. The fine structures in the ground truth are clearly smoothed, limited by the spatial resolution that is inherent to the Level 2 pixel sizes (10 km). However, by oversampling at a fine grid (0.2 km for the first row vs. 5 km for the second row), the spatial gradients are better recovered, and spatial features finer than individual Level 2 pixels can be identified. Additionally, the details in the spatial response function is better resolved with a finer target grid, which is particularly beneficial when collocating with higher resolution measurements (e.g., a cloud imager). As such, although the spatial resolving power is ultimately determined by the spatial extent of satellite pixels, the physical oversampling approach helps enhancing the visualization of spatial gradient and the identification of emission sources."**

Figure 8 of the revised manuscript:

[Figure]

**Figure 8. First column: spatial response function of an imaginary sensor discretized at 0.2 km (top) and 5 km (bottom) grid. Second column: ground truth spatial distribution generated as an array of 2-D Gaussian functions of same height (the top and bottom panels are the same). The FWHM of each Gaussian is labeled. Third column: physical oversampling results using 10000 randomly generated observations and discretized at 0.2 km (top) and 5 km (bottom) grid. The pixel size, which determines the spatial resolution, is labeled as the inserted white boxes.**

*In summary, besides the application just mentioned, it is unclear whether the much higher-than-pixel resolution output can be justified. In the past literature, Fioletov et al. 2011 (GRL) and Fioletov et al. 2013 (JGR:A) claims 'detailed "subpixel-resolution" spatial distribution' is possible and Streets et al. 2013 (Atmos. Environ.) refers to oversampling as achieving "super-resolution". To what extent is the enhanced resolution (such as 1 km, as proposed in this manuscript) physically real? I suggest adding discussion of the limitations of the approach and caveats in interpreting the results of oversampling. This will strengthen the manuscript and help the community who may use the described algorithm or other similar methods in the future.*

The advantage of fine-grid output vs. coarse grid output can be easily seen by comparing the first row (0.2-km grid) with the second row (5-km grid) in Fig. 8 of the revised manuscript (this figure is included in the response to the previous comment) and by comparing the first column (10-km grid) with columns 2-4 (1-km grid) in Fig. 8 of the original manuscript (now Fig. 9 in the revised manuscript). See responses to previous comments for clarification of spatial resolution vs. spatial sampling.

*A point on the nomenclature of oversampling: while the term has been used in some past satellite papers, nevertheless I find it problematic since it is quite different than how it used in signal processing, where there is a well-known and widely used meaning of natively sampling at high resolution and then converting to a lower one. The authors note instead that the presented algorithm is a type of interpolation. I believe that referring to the algorithm as 'gridding' rather than 'oversampling' is formally correct and gives a much better intuition of what the algorithm actually does.*

As shown in the response to the first comment, we believe oversampling is the appropriate nomenclature.

*A minor point - 'agile' is also used to describe the algorithm, can a few words be added to clarify what the intended meaning is?*

It means that the algorithm works for different satellite sensors with quadrilateral/elliptical pixel shapes and different pixel sizes, and that the sensitivity distribution can be flexibly chosen by changing the parameters of 2-D super Gaussian function. See page 3, line 7 of the manuscript.

"**In this work, we present an agile, physics-based oversampling approach that represents each Level 2 satellite pixel as a sensitivity distribution on the Earth's surface (e.g., the spatial response function), instead of a point or a polygon as assumed in previous methods.**"

References:

1.  Chance, K., Kurosu, T. P. & Sioris, C. E. Undersampling correction for array detector-based satellite spectrometers. *Appl. Opt.* **44,** 1296–1304 (2005).

2.  de Foy, B., Lu, Z., Streets, D. G., Lamsal, L. N. & Duncan, B. N. Estimates of power plant NOx emissions and lifetimes from OMI NO2 satellite retrievals. *Atmos. Environ.* **116,** 1–11 (2015).

3.  Liu, F. *et al.* NOx lifetimes and emissions of cities and power plants in polluted background estimated by satellite observations. *Atmos. Chem. Phys.* **16,** 5283–5298 (2016).

4.  Valin, L. C., Russell, A. R. & Cohen, R. C. Variations of OH radical in an urban plume inferred from NO2 column measurements. *Geophys. Res. Lett.* **40,** 1856–1860 (2013).

5.  Beirle, S., Boersma, K. F., Platt, U., Lawrence, M. G. & Wagner, T. Megacity emissions and lifetimes of nitrogen oxides probed from space. *Science* **333,** 1737–1739 (2011).

---

## Author Comment (AC3) · 14 Nov 2018

Response to Referee #2:

We appreciate the very helpful feedback from the referee. The referee's comments are listed in *italics*, followed by our response in blue. New/modified text in the manuscript is in **bold**.

*1. The definition of total number of overlapping pixel polygons used in averaging for grid cell j (formula (4)) is somewhat confusing. To my understanding, S(i, j) is the overlapping area, thus $D(j) = \sum_i S(i,j)$ is just sum of overlapping area and the unit of $D(j) = \sum_i S(i,j)$ is km². If so, is the unit of D(j) for tessellation in figure 8 and 9 also km2? Is D(j) actually defined as $\sum_i \frac{S(i,j)}{\sum_j S(i,j)}$ or $\sum_i \frac{S(i,j)}{grid\ cell\ area}$ in figure 8 and 9?*

The scaling of *S(i,j)* does not matter as it appears both in *A* and *B* and will be normalized out when calculating *C* (Eq. 1-3). To clarify *S(i,j)*, we always define it as the fractional overlapping area (a dimensionless number). As such, *D(j)* is always dimensionless (sum of fractional overlapping area), and can be understood as the number of overlapping pixels for grid cell *j*. This is what was presented in Fig. 8 and 9. We clarified the sentences at page 7, lines 13-14 as:

**"When the destination grid is regular with constant grid cell area, it is convenient to normalize S(i,j) by the grid cell area, leading to overlapping fractions. We will follow this convention hereafter, and hence S(i,j) is always a dimensionless number".**

*2. In formula (10), why $\int\int_{grid\ j} S(x,y|i)dxdy$ is normalized by grid cell area? If S(i, j) is just defined as $\int\int_{grid\ j} S(x,y|i)dxdy$, $W(i,j) = \frac{S(i,j)}{\sum_j S(i,j)}$ is the normalized spatial response function for observation i and its spatial integration $\sum_j W(i,j)$ is unity. Considering discretization of spatial response function in computation, both definitions of S(i, j) are fine. Physically, should $\int\int_{grid\ j} S(x,y|i)dxdy$ be normalized by grid cell area or not?*

As indicated by the referee, whether normalizing *S(i,j)* by the grid cell area or not does not change the oversampling results. We choose to normalize the spatial integral of the spatial response function *S(x,y)* by the grid cell area, because (1) the resultant *S(i,j)* is compatible with the definition in tessellation (see the response to the previous comment) and (2) the spatial integration of *S(x,y)* over grid *j* can be directly approximated by *S(x,y)* evaluated at the center of grid *j*. In this way, *S(i,j)* is always a dimensionless number between 0 and 1. To clarify, one sentence is added to page 11, line 7:

**"Similarly to the tessellation approach, S(i,j) is always a dimensionless number between 0 and 1."**

*3. Levi Golston has a good point of what is the extent of the enhanced resolution result physical real (Short comment 1 for this discussion paper, https://www.atmosmeas-tech-discuss.net/amt-2018-253/amt-2018-253-SC1-supplement.pdf). Levi Golston shows an example that "true" value is unity while oversampling result is 1/63. The result is, however, based on using 2-D boxcar spatial response function on observation generation and oversampling. If 2-D super Gaussian*

*function is used, will it show better oversampling result? For 2-D super Gaussian function, will small k1 and k2 give better result than larger ones for Levi Golston's example? I suggest adding discussion of it.*

Strictly speaking, the spatial resolution is only determined by the satellite (e.g., we say TROPOMI has a higher resolution than OMI). We may choose a very fine target grid (i.e., oversample the spatial distribution) but that does not help reproducing the true concentration distribution once the satellite spatial response is adequately resolved. To avoid confusion, we replaced the term "**grid resolution**" in the manuscript by "**grid size**".

If the true spatial response function of the sensor is used in the physical oversampling, and the spatial response is accurately integrated over each grid (Eq. 10 of the manuscript), the result is always physically real as it represents the "exact" observation of the sensor. However, the exact observation does not equal to the true concentration distribution due to the limitation of satellite pixel sizes. One subsection (section 4.3) and one figure (Fig. 8 of the revised manuscript) is added to clarify the definition of spatial resolution vs. spatial sampling:

"**The difference between resolution and sampling density for 1-D spectral data has been thoroughly discussed in the literature (e.g., Chance et al., 2005). However, for 2-D, spatially resolved data, it is common to refer to both the sizes of the Level 2 pixels and the size of the Level 3 grid as the spatial "resolution" of the data. To avoid confusion, it is emphasized here that the true spatial resolution is limited by the sizes of Level 2 pixels. The size of Level 3 grid only determines the density of spatial sampling, which does little to enhance the true resolving power of the data after reaching a certain point. For example, the oversampling results using synthetic OMI data at 1 km vs. 0.05 km grids are very similar (Fig. 6). Nonetheless, it is still beneficial to oversample, i.e., make Level 3 grid size significantly smaller than Level 2 pixel sizes, as demonstrated by Fig. 8. As the ground truth, an array of 2-D Gaussian functions are generated with FWHM ranging from 1 km to 16 km (the second column of Fig. 8) and peak height of unity, and this true field of concentration is measured by an imaginary sensor whose spatial response function is a 2-D super Gaussian (Eq. 8) with FWHM = 10 km and $k_1 = k_2 = 8$ (the first column and the white boxes inserted in the third column). The third column shows the oversampling results using 10000 randomly located observations. The fine structures in the ground truth are clearly smoothed, limited by the spatial resolution that is inherent to the Level 2 pixel sizes (10 km). However, by oversampling at a fine grid (0.2 km for the first row vs. 5 km for the second row), the spatial gradients are better recovered, and spatial features finer than individual Level 2 pixels can be identified. Additionally, the details in the spatial response function is better resolved with a finer target grid, which is particularly beneficial when collocating with higher resolution measurements (e.g., a cloud imager). As such, although the spatial resolving power is ultimately determined by the spatial extent of satellite pixels, the physical oversampling approach helps enhancing the visualization of spatial gradient and the identification of emission sources.**"

Figure 8 of the revised manuscript:

[Figure]

**Figure 8.** First column: spatial response function of an imaginary sensor discretized at 0.2 km (top) and 5 km (bottom) grid. Second column: ground truth spatial distribution generated as an array of 2-D Gaussian functions of same height (the top and bottom panels are the same). The FWHM of each Gaussian is labeled. Third column: physical oversampling results using 10000 randomly generated observations and discretized at 0.2 km (top) and 5 km (bottom) grid. The pixel size, which determines the spatial resolution, is labeled as the inserted white boxes.

---

## Referee Report (RR1)

Thank you for making these modifications. It may be helpful to include two additional points in the abstract, which have already been made in Section 5.1:

1. The physical oversampling method is especially advantageous during smaller temporal windows (such as 5 days' worth of data). As noted by the authors in section 5.1, the results from other oversampling techniques become increasingly similar for longer averaging times.
2. There is no appreciable difference in the computational time when using the physical oversampling method versus other oversampling methods.

Also, please make your code of the physical oversampling method available to the public.

---

## Author Response (AR2)

Dear Dr. Wang:

We have included the following sentences in the abstract based on suggestions from the referees:

"This physical oversampling approach is especially advantageous during smaller temporal windows and shows substantially improved visualization of trace gas distribution and local gradients when applied to OMI NO2 products and IASI NH3 products. There is no appreciable difference in the computational time when using the physical oversampling versus other oversampling methods."

In addition, the code of physical oversampling is already publicly available. A Code availability section is included after the conclusions:

"A MATLAB implementation of the physical oversampling is available at <a href="https://github.com/Kang-Sun-CfA/Oversampling\_matlab/">https://github.com/Kang-Sun-CfA/Oversampling\_matlab/</a>."

Best regards,

Kang Sun

**A physics-based approach to oversample multi-satellite, multi-species observations to a common grid**

Kang Sun1, Lei Zhu2, Karen Cady-Pereira3, Christopher Chan Miller4, Kelly Chance4, Lieven Clarisse5, Pierre-François Coheur5, Gonzalo González Abad4, Guanyu Huang6, Xiong Liu4, Martin Van Damme5, Kai Yang7, and Mark Zondlo8

1Research and Education in eNergy, Environment and Water Institute, University at Buffalo, Buffalo, NY, USA

2School of Engineering and Applied Sciences, Harvard University, Cambridge, MA, USA

3Atmospheric and Environmental Research, Lexington, MA, USA

4Harvard-Smithsonian Center for Astrophysics, Cambridge, MA, USA

[revised manuscript text omitted]